# Test-Time Defense Against Adversarial Attacks via Stochastic Resonance of Latent Ensembles

## Abstract

We propose a test-time defense mechanism against adversarial attacks: imperceptible image perturbations that significantly alter the predictions of a model. Unlike existing methods that rely on feature filtering or smoothing, which can lead to information loss, we propose to "combat noise with noise" by leveraging stochastic resonance to enhance robustness while minimizing information loss. Our approach introduces small translational perturbations to the input image, aligns the transformed feature embeddings, and aggregates them before mapping back to the original reference image. This can be expressed in a closed-form formula, which can be deployed on diverse existing network architectures without introducing additional network modules or fine-tuning for specific attack types. The resulting method is entirely training-free, architecture-agnostic, and attack-agnostic. Empirical results show state-of-the-art robustness on image classification and, for the first time, establish a generic test-time defense for dense prediction tasks, including stereo matching and optical flow, highlighting the method's versatility and practicality. Specifically, relative to clean (unperturbed) performance, our method recovers up to 68.1% of the accuracy loss on image classification, 71.9% on stereo matching, and 29.2% on optical flow under various types of adversarial attacks.

## 1 Introduction

Most deep neural networks in use today are deterministic maps from a fixed-size input to a fixed-size feature vector. In auto-regressive Transformer models, that vector encodes the next element (token) in the input sequence. Similarly, in convolutional architectures, that vector may encode the input data. In either case, the output vector is often highly sensitive to perturbations of the input, and one can intentionally choose these imperceptible perturbations *adversarially* so as to maximize the change in the output Goodfellow et al. (2014). In some cases, the same perturbation can even be disruptive for a large number of possible inputs Moosavi-Dezfooli et al. (2017), exploiting the convoluted geometry of the decision boundary imposed by such trained models Tramèr et al. (2017). This spurious sensitivity could be exploited adversarially to disrupt the operation of a model.

From a classical perspective of signal processing, adversarial perturbations of images appear as small high-frequency "noise" resembling *aliasing* artifacts. These are imperceptible since the human visual system easily discounts them on account of their poor fit to the 'ecological statistics' of natural images Gibson (2014). Classical sampling theory prescribes anti-aliasing by low-pass filtering the data, removing information along with the artifacts. Low-pass filtering consists of spatial averaging of the perturbed data with respect to a chosen kernel, typically a Gaussian or a constant ("pillbox"). Alternatively, one could "denoise" the embeddings by averaging output vectors, also a lossy operation. The choice of the kernel should match the statistics of the perturbations, which sets up a cat-and-mouse game where the adversary can easily modify the perturbations to bypass the anti-aliasing filter, namely *adaptive* attacks Tramer et al. (2020), and the model needs to constantly be fine-tuned to "anti-alias" new forms of adversarial perturbations. In the context of deep neural networks, existing defense methods that manipulate feature representations Xie et al. (2019); Bai et al. (2021); Yan et al. (2021); Kim et al. (2023) fundamentally adhere to this paradigm in principle. Despite substantial engineering efforts, these methods remain inherently vulnerable to adaptive attacks because they rely on pre-defined filtering strategies that are fixed at inference time.

**Desiderata:** To break this cycle, we advocate a method to mitigate the effect of adversarial perturbation that (i) operates at test time, without the need to update the model weights, and (ii) does not entail information loss associated with direct or indirect spatial filtering. In addition, it would be ideal if this method could (iii) be applied to existing network architectures without modifications, and (iv) be agnostic to the specific type of adversarial perturbation.

**Stochastic resonance** is a technique that resolves a quantized signal below the quantization level, by quantizing and ensembling perturbed versions of the signal Benzi et al. (1981). It has been used extensively in cochlear implants, where power constraints limit the resolution of the digital circuitry Stocks et al. (2002). It has also been used to 'super-resolve' Vision Transformer embeddings, where entire patches are encoded into a vector, which is computed at a coarsely subsampled grid Lao et al. (2024). In this paper, instead, we use Stochastic Resonance for the opposite purpose, *not* to super-resolve the quantized signal, but to perform latent ensembling to remove the effects of adversarial perturbations in the embedding.

Rather than averaging the data, or averaging their embedding as in classical denoising, *we average transformed embeddings* in latent space. This averaging is performed over small transformations sampled at random or deterministically from the group of planar translations, by computing the encoding of the transformed image, and then mapping the encoding back through the push-forward of the inverse transformation. This process can be expressed as a single formula in equation 1. Since the embedding is typically computed on a coarse grid, but the transformations are sampled on the native lattice of the image, the resulting embedding is free of spatial averaging artifacts. As with other uses of Stochastic Resonance, the effect is seemingly paradoxical as *we combat noise with noise*: We apply purposeful perturbations to eliminate the effect of adversarial perturbations.

Our method can be thought of as marginalizing the translation group in latent space with respect to a chosen prior, which is the only design choice in our method. We choose the simplest, which is the constant prior. The purposeful perturbations alter the spatial sampling, and the implicit ensembling in latent space averages out the effect of sampling artifacts, thwarting the effect of adversarial perturbations. It is as if we were given multiple images with different 'noise', except that the noise in question is not the adversarial perturbation, but the splinters of adversarial perturbations obtained by different spatially quantized versions of the perturbed image, due to the translational perturbations, which are then averaged out by the latent ensembling.

**Outcomes:** Our method fulfills the desiderata (i)-(iv) laid out earlier: (i) It does not require training or fine-tuning; (ii) it minimizes information loss by latent ensembling of perturbed embeddings; (iii) it can be applied to different network architectures and tasks, *including* networks already equipped with different defense techniques like adversarial training, and (iv) is agnostic to the specific perturbation. To measure the effectiveness of our method in mitigating the effects of adversarial perturbations, we test it on three vastly different tasks, including image classification, and two other dense prediction tasks: stereo matching and optical flow. where we are the first to show a significant and consistent improvement in robustness to various adversarial attacks.

One could argue that there is still a cat-and-mouse game in our setting, if the adversary knows our technique and tailors the adversarial perturbations to bypass it. To assess this risk, we conduct "worst-case" adaptive tests to measure the performance of our method under adaptive attacks when the attacker knows the exact defense strategy, thus the adversarial perturbation is designed to maximally disrupt the result end-to-end, *including* our stochastic resonance. Our results show that the method is resistant to breaking even when the adversary optimizes adaptively through it end-to-end.

## 2 RELATED WORK

The literature on adversarial attack and defense is extensive. We highlight some of the advances.

**Adversarial Training as Defense.** Adversarial training increases the robustness of the model by training it with adversarially augmented images. The popular attack methods used are Fast Gradient Sign Method (FGSM) Goodfellow et al. (2014), and Projected Gradient Descent (PGD) Madry et al. (2017). ALP Kannan et al. (2018) Minimizes the difference between the logits of pairs of clean and adversarially augmented images. TRADES Zhang et al. (2019) decomposes prediction error for adversarial examples into natural error and boundary error to improve adversarial robustness. MART Wang et al. (2019) improves adversarial robustness by considering misclassified natural examples

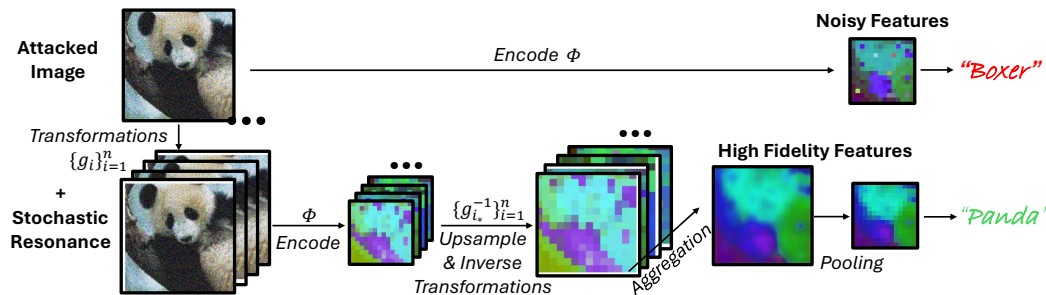

Figure 1: **Defense against adversarial attacks via stochastic resonance.** *Neural networks are sensitive to perturbations in the input space, which adversarial attacks exploit to manipulate outputs. Conventional defense primarily focuses on filtering out unreliable features or denoising either the input or the features. Instead of removing noise, we propose a novel defense by introducing noise. Based on stochastic resonance, controlled transformations are introduced to the input. Features are then aggregated after inverting these transformations. The resulting test-time defense method requires no training and is compatible with diverse architectures. Notably, it not only improves robustness against adversarial attacks but also increases the difficulty of crafting adversarial examples, even when the attacker is aware of whether and how stochastic resonance is being used.*

during training. Subsequent work Cai et al. (2018); Zhang et al. (2020); Wang & Wang (2022); Jin et al. (2022); Ghiasvand et al. (2024); Xu et al. (2024); Gopal et al. (2025) improves adversarial robustness with curriculum learning Bengio et al. (2009), model ensembling, critical parameter selection, and gradient tracking. On the other hand, some methods learn robust feature representation through a modified architecture or feature manipulation. Galloway et al. (2019); Benz et al. (2021); Wang et al. (2022) investigate the effect of batch normalization on adversarial robustness. Dhillon et al. (2018); Madaan et al. (2020) prunes certain activations in the network that are susceptible to adversarial attacks. Xiao et al. (2019) keep k-features with the largest magnitude and deactivate everything else. Zoran et al. (2020) uses an attention mask to highlight robust regions on the feature. Feature Denoising (FD) Xie et al. (2019) uses classical denoising techniques to deactivate abnormal activations. Bai et al. (2021); Yan et al. (2021) proposed Channel Activation Suppression (CAS) and Channel-wise Importance-based Feature Selection (CIFS) to deactivate feature channels that are vulnerable to attacks. Kim et al. (2023) improves the robustness with Feature Separation and Recalibration (FSR). Our method also operates in feature space, but purely at test time. While we choose some as baselines, our method can work in conjunction with any aforementioned methods.

**Adversarial Purification as Defense** Another line of work focuses on purifying or augmenting the images before they are used as input. Tang & Zhang (2024); Yeh et al. (2024); Tsai et al. (2023) train an FGSM robust classifier, a diffusion model, or a mask auto-encoder, respectively, to purify adversarial examples. Wang et al. (2021) optimizes both the model and the input to minimize the entropy of model predictions to adapt to changing attacks. Cohen & Giryes (2024) trains a random forest predictor to ensemble outputs from test-time augmented images. These works involve training a new model or updating the original model, while our method is purely test-time and does not require any training. Pérez et al. (2021) ensembles model output from different augmentations, which is a special case of our method, as the ensemble is performed solely on the output, while we can ensemble at any layer, which both saves computational cost and achieves higher performance. Notably, we recognize that above methods focus solely on classification as a task for adversarial attacks. Through extensive experiments, we demonstrate that our method can not only perform well on classification, but also on dense prediction tasks like optical flow, and stereo matching.

**Stochastic Resonance (SR)** was proposed by Benzi et al. (1981) and first applied in climate dynamics (Benzi et al., 1982) and later in signal processing (Wellens et al., 2003; Kosko & Mitaim, 2001; Chen et al., 2007) and acoustics (Shu-Yao et al., 2016; Wang et al., 2014). While several works Zhang et al. (2022); Ukita & Ohki (2023) introduce stochasticity within feature space, the most closely related prior effort is Lao et al. (2024), which applies stochastic resonance to "super-resolve" Vision Transformer (ViT) embeddings. In this work, we instead use SR to mitigate adversarial attacks. Since SR has been developed specifically to address *quantization artifacts*, it has never before been used to mitigate classes of perturbations beyond aliasing. Our novel use of the technique leverages the fact that group transformations and spatial quantization preserve the statistics of natural images, which are heavy-tailed, but do not preserve the statistics of adversarial perturbations.

## 3 METHOD

**Notation.** A digital image $x \in [0, \; L-1]^{W \times H}$ can be described as a map from a discrete planar lattice $\Lambda \subset \mathbb{R}^2$ with $H$ rows and $W$ columns to $L$ discrete levels, $x : \Lambda \to [0, \; L-1]$; a 'feature' or 'embedding' of an image $x$ is the output of an encoder $\phi$ that maps it to a vector space with $K$ channels, typically through a parametric trained model:

$$\phi : x \mapsto \phi(x) \in \mathbb{R}^K.$$

We represent a group transformation of the image through an operator $g : \mathbb{R}^2 \to \mathbb{R}^2$, which can be restricted to the lattice $\Lambda$ through padding, sampling and quantization at the expense of invertibility:

$$g : x \mapsto g(x) \in \Lambda \subset \mathbb{R}^2.$$

For example, a translation by an integer pixel can be represented by an upper diagonal matrix $G$, $g(x) = Gx$ with ones above the diagonal. The group $g$ operating on $x$ induces an operation on $\phi$ via

$$g_* \phi(x) \doteq \phi(g(x)).$$

We call the composition of $\phi$ and $g$ the encoding of the transformed image

$$\psi(x) \doteq \phi(g(x)) = \phi \circ g(x).$$

The main object of interest in our method is:

$$g_*^{-1} \psi(x) = g_*^{-1} \circ \phi \circ g(x).$$

This is obtained, reading right-to-left, by first transforming the image, then passing it through an encoder, and then transforming back the feature map through the push-forward action $g_*^{-1}$.

**Perturbations.** We consider two types of perturbations, extraneous and purposeful. The extraneous one could be an additive perturbation to an image, $\tilde{x} = x + n$, designed to maximally change the value of the embedding (adversarial perturbation) $\phi(\tilde{x})$:

$$\tilde{x} = x + n(x) \quad | \quad n(x) = \arg\max d(\phi(x), \phi(\tilde{x})), \; |n| < \epsilon$$

for some small $\epsilon$ designed so the perturbation is, ideally, imperceptible by humans.

The purposeful perturbations are small group actions $g_i, \; i = 1, \ldots, N$, which could be sampled deterministically or at random according to some chosen distribution $g_i \sim P_g$, either way yielding a set $\{g_1(x), \ldots, g_N(x)\}$. Our goal is to use these purposeful perturbations to combat the effects of extraneous adversarial perturbations.

**Averaging, smoothing, and stochastic resonance.** The resemblance between adversarial perturbations and aliasing artifacts has motivated the use of anti-aliasing, or smoothing, techniques to mitigate them. These consist of *spatial averaging* of the data prior to computing the map $\phi$. If we call $B_{ij}^\sigma$ a neighborhood of size $\sigma > 0$ around $(i, j) \in \Lambda$,

$$B_{ij}^\sigma = \{(i', j') \in \Lambda \mid d((i, j), (i', j')) \le \sigma\}$$

then the simplest form of smoothing is simply averaging in a neighborhood,

$$\bar{x}_{i,j} = \frac{1}{\sigma^2} \sum_{(i', j') \in B_{i,j}^\sigma} x_{i'j'}$$

which we write in terms of translations $g(x_{i,j}) = x_{i+u,j+v}$ within the same neighborhood $B^\sigma$,

$$\bar{x} = \frac{1}{N} \sum_{i=1}^N g_i(x).$$

One can then obtain an encoding by smoothing the embedding

$$\bar{\phi}(x) = \frac{1}{N} \sum_{i=1}^N \phi(g_i(x)).$$

This can be interpreted as marginalizing the translation group with a prior $P_g$ when computing $\phi$. Notice that the average can be computed on a coarser lattice $\tilde{\Lambda}$, but its value still depends on data

in the finer grid $\Lambda$. Classical Sampling Theory teaches that smoothing mitigates the effect of high-frequency aliasing $n$ at the cost of information loss on $x$.

Stochastic resonance also marginalizes the translation group, but by *averaging transformed data in latent space:*

$$\hat{\phi}(x) = \frac{1}{N} \sum_{i=1}^{N} g_{i_*}^{-1} \circ \phi \circ g_i(x). \tag{1}$$

More general groups, and more general averaging kernels, can be considered although we find that the simplest case described here already suffices.

**Stochastic resonance as a defense against adversarial attacks.** To analyze the effect of an adversarial perturbation, consider a first-order expansion at each transformed input:

$$\phi(g_i \tilde{x}) = \phi(g_i x + g_i n(x)) \approx \phi(g_i x) + J_{g_i x}[g_i n(x)],$$

where $J_{g_i x}$ is the Jacobian of $\phi$ at $g_i x$. Substituting with equation 1 yields the perturbation term

$$\Delta_{\mathrm{SR}}(x) = \hat{\phi}(\tilde{x}) - \hat{\phi}(x) \approx \frac{1}{N} \sum_{i=1}^{N} g_{i_*}^{-1} J_{g_i x}[g_i n(x)].$$

By contrast, a standard feedforward encoder responds as

$$\Delta_{\mathrm{base}}(x) = \phi(\tilde{x}) - \phi(x) \approx J_x \, n(x),$$

and is therefore dominated by the local worst-case Lipschitz direction at $x$.

The terms $g_{i_*}^{-1} J_{g_i x}[g_i n(x)]$ in the stochastic resonance expansion are generally *not aligned* across $i$, for several reasons. First, the adversarial noise $n(x)$ is constructed for the quantization pattern and receptive field alignment at a *single* input $x$; under translation, it is re-aliased into different patches. Second, the Jacobian $J_{g_i x}$ changes with $i$, so the maximally amplifying direction at $x$ is not simultaneously maximally amplifying at all transformed inputs. Third, after inverse warping by $g_{i_*}^{-1}$, the signal terms $\phi(g_i x)$ align and add coherently, whereas the Jacobian–noise terms do not.

Under mild assumptions (e.g., approximate decorrelation across $i$),

$$\left\| \frac{1}{N} \sum_{i=1}^{N} v_i \right\|^2 \sim \frac{1}{N} \mathbb{E} \|v_i\|^2, \qquad v_i = g_{i_*}^{-1} J_{g_i x}[g_i n(x)],$$

so the effective Lipschitz constant is reduced by a factor $1/\sqrt{N}$ by stochastic resonance, which suppresses adversarially amplified latent-space outliers while preserving scene-consistent features.

**Purposeful perturbation.** The only design choice in the method is the set of purposeful perturbations. While that can be optimized for performance, we optimize for simplicity, restricting our attention to translation by integer pixels. We know that, for adversarial perturbations, $d(\phi(\tilde{x}), \phi(x))$ is large, where $d(\cdot)$ defines the distance between features. Ideally, for stochastic resonance, we want $d(\hat{\phi}(\tilde{x}), \hat{\phi}(x)) = 0$ while keeping $\hat{\phi}$ as information-preserving as possible. The theory of Stochastic Resonance shows that, if we sub-sample a signal from its native granularity $\Lambda$ to a coarser grid $\tilde{\Lambda} \subset \Lambda$, and choose the purposeful perturbations to act on the finer grid $\Lambda$, under certain conditions one can recover the original signal at the finer granularity Benzi et al. (1981). In this paper, we focus on testing whether $\hat{\phi}$ is insensitive to adversarial perturbations. We do so empirically in Sect. 4.

## 4 EXPERIMENTS

While $g$ can be sampled from any invertible group transformation (e.g., rotation, scaling), we implement stochastic resonance using integer-pixel translations, denoted as $\{g_i\} = \{(x, y) | x \in [-d_x, d_x], j \in [-d_y, d_y]\}$, following the approach in SRT Lao et al. (2024), which avoids interpolation artifacts. While the networks' inherent sensitivity to pixel-level shifts is typically regarded as detrimental due to the "flickering problem" Azulay & Weiss (2019); Sundaramoorthi & Wang (2019), our approach, on the contrary, leverages it to defend against adversarial perturbations. Given these perturbations $\{g_i\}$, ensembling can be performed at any chosen layer of the network. Features are aggregated as described in Eq. 1 and then passed to downstream network components.

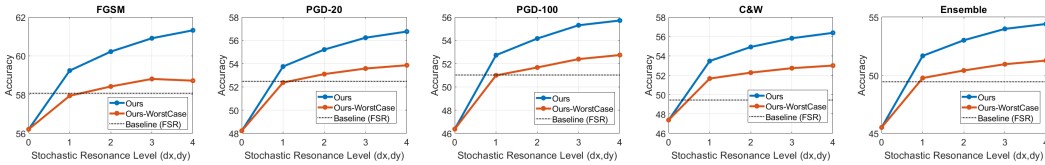

Figure 2: **Results on CIFAR-10 under varying levels of stochastic resonance.** *Increasing the stochastic resonance level consistently enhances robustness across all settings, yielding clear gains over the baseline method (FSR). Notably, our approach achieves superior performance even under adaptive adversarial attacks (Ours-WorstCase), despite the baseline being evaluated only in the non-adaptive case.*

| Method | Architecture | Natural | FGSM | PGD-20 | PGD-100 | C&W | Ensemble | AutoAttack |
|---|---|---|---|---|---|---|---|---|
| AT | | 85.02 | 56.21 | 48.22 | 46.37 | 47.38 | 45.90 | 44.11 |
| + FD | | 85.14 | 56.81 | 48.54 | 46.70 | 47.72 | 45.82 | 44.57 |
| + CAS | | **85.78** | 55.57 | 50.42 | 49.91 | 53.47 | 46.46 | 44.23 |
| + CIFS | ResNet-18 | 79.87 | 56.53 | 49.80 | 48.17 | 49.89 | 47.26 | 43.94 |
| + FSR | | 81.46 | 58.07 | 52.47 | 51.02 | 49.44 | 48.34 | 46.41 |
| + TTE | | 85.25 | 59.20 | 53.00 | 51.65 | 52.45 | 50.60 | - |
| + Ours | | 84.93 | **61.02** | **56.08** | **55.17** | **55.53** | **53.68** | **49.78** |
| + Ours-WorstCase | | 84.93 | 58.81 | 53.58 | 52.39 | 52.73 | 50.95 | - |
| TRADES | WideResNet-34 | 84.92 | 60.87 | 56.13 | 55.16 | 54.02 | 53.38 | 46.81 |
| +Ours | | **85.03** | **62.43** | **58.64** | **57.87** | **57.18** | **56.28** | **51.80** |
| MART | ResNet-18 | **83.07** | 60.21 | 54.14 | 52.90 | 49.62 | 48.95 | 44.27 |
| + Ours | | 82.70 | **62.62** | **59.03** | **58.13** | **55.51** | **54.61** | **50.72** |

Table 1: **Defense against adversarial attacks on classification task (CIFAR-10).** *Compared to baselines that filter or manipulate features, ours does not modify network architecture or weights. Instead, ours performs an ensemble in the feature space. On CIFAR-10, ours achieves state-of-the-art robustness without requiring any additional training. Moreover, even in a worst-case adaptive adversary setting where the attacker is fully aware of the defense and how stochastic resonance is applied, the effectiveness of adversarial attacks is still notably reduced, while the computational cost for executing such attacks is significantly increased.*

## 4.1 IMAGE CLASSIFICATION

**CIFAR-10.** We evaluate on CIFAR-10 Krizhevsky et al. (2009), building upon the standard and publicly available code base of FSR Kim et al. (2023) and accompanying evaluation protocol (e.g. AutoAttack Croce & Hein (2020)). We apply stochastic resonance to networks pre-trained with AT Madry et al. (2017), TRADES Zhang et al. (2019), and MART Wang et al. (2019), using their publicly released weights. Our method operates with these methods purely at test time. We conduct experiments on ResNet-18 He et al. (2016) and WideResNet-34 Zagoruyko & Komodakis (2016), depending on the availability of author-released pre-trained weights. In all cases, feature ensembling is performed before the final linear layer. Furthermore, we consider a worst-case adaptive adversary setting, where the attacker has full knowledge of the model weights, and knows every stochastic resonance transformation by accessing *every single* forward pass and its gradients.

Fig. 2 shows the results varying different levels of stochastic resonance. Increasing the resonance level consistently enhances robustness in all attack settings, leading to substantial improvements over the baseline method (FSR). Importantly, our approach surpasses the baseline even under adaptive adversarial attacks. Additionally, we compare our method against multiple baselines, including feature-level manipulation methods (FD Xie et al. (2019), CAS Bai et al. (2021), CIFS Yan et al. (2021)) and ensemble-based approach TTE Pérez et al. (2021). The results, summarized in Tab. 1, demonstrate that stochastic resonance consistently outperforms all baselines across different attacks. Even in the worst-case adaptive attack scenario, where the adversary accounts for all stochastic resonance forward passes, the model remains significantly more robust than the baseline methods. In addition, stochastic resonance increases attack complexity under the adaptive settings, making adversarial noise generation more challenging for the attacker. As a result, the computational cost for generating adaptive adversarial perturbations increases substantially. For example, with stochastic resonance, 8x more wall-clock time is required to create adversarial examples with PGD-100.

**ImageNet.** We further evaluate our approach on the ImageNet Deng et al. (2009) classification dataset using standard segmentation backbones, including ResNet-50 He et al. (2016) and Vision

| Att. Strength ($\epsilon$) | 8/255 | | 4/255 | | 2/255 | | 1/255 | |
|---|---|---|---|---|---|---|---|---|
| Metric | Top-1 | Top-5 | Top-1 | Top-5 | Top-1 | Top-5 | Top-1 | Top-5 |
| No Defense | 4.51 | 19.55 | 12.25 | 42.15 | 29.79 | 65.09 | 48.09 | 78.13 |
| Ours ($d=1$) | 11.66 | 46.27 | 27.85 | 65.80 | 45.94 | 77.63 | 57.48 | 83.43 |
| Ours ($d=2$) | 18.78 | 58.58 | 36.42 | 72.86 | 51.58 | 80.85 | 60.41 | 84.77 |
| Ours ($d=3$) | **25.77** | **65.88** | **42.52** | **76.75** | **54.94** | **82.64** | **62.08** | **85.50** |

Table 2: **ImageNet with ViT-Small.** Increasing the level of stochastic resonance consistently improves both Top-1 and Top-5 accuracy under adversarial attacks. Relative to the clean baseline (72.9 Top-1, 92.91 Top-5), our method recovers up to 55.8% of the Top-1 accuracy loss and 68.1% of the Top-5 accuracy loss.

| Att. Strength ($\epsilon$) | 8/255 | | 4/255 | | 2/255 | | 1/255 | |
|---|---|---|---|---|---|---|---|---|
| Metric | Top-1 | Top-5 | Top-1 | Top-5 | Top-1 | Top-5 | Top-1 | Top-5 |
| No Defense | 8.51 | 26.19 | 9.45 | 27.92 | 11.10 | 30.63 | 15.21 | 36.76 |
| Initial Conv. | 16.91 | 34.66 | 17.36 | 35.28 | 18.12 | 36.30 | 19.59 | 38.25 |
| Res. Block 1 | 22.44 | 44.95 | 23.39 | 46.03 | 24.84 | 47.64 | 27.57 | 50.80 |
| Res. Block 2 | 24.88 | 48.54 | 25.90 | 49.66 | 27.23 | 51.19 | 30.11 | 54.15 |
| Res. Block 3 | 21.76 | 44.28 | 22.73 | 45.47 | 24.14 | 47.14 | 27.18 | 50.20 |

Table 3: **Layer-wise ablation on ResNet-50.** Adversarial perturbations resemble high-frequency noise. Applying stochastic resonance through shallow layers is sufficient to defend against adversarial attacks, substantially reducing the overall computational cost.

Transformer Dosovitskiy et al. (2020), without adversarial training. As in the CIFAR experiments, we vary the level of stochastic resonance and conduct ablation studies by testing against PGD attacks of different strengths. Tab. 2 reports the results for ViT-Small. Consistent with the CIFAR-10 findings, increasing the resonance level leads to consistent improvements in robustness, as measured by both Top-1 and Top-5 accuracy. Notably, the vanilla ViT-Small model without attack achieves 72.9 (Top-1) and 92.91 (Top-5), which means our method recovers the accuracy drop under adversarial attacks by up to a relative 55.8% (Top-1, when $\epsilon = 4/255$) and 68.1% (Top-5, when $\epsilon = 2/255$). We further evaluate our method on ResNet-50 and observe a consistent trend, as shown in Tab. 4.

We also explored group transformations other than translation, e.g. rotations in Tab. 4. For a fair comparison, we use the same number of augmentations as in the translation experiments. Rotations behave similarly to translations at low levels of stochastic resonance, but performance degrades as the resonance level increases. We hypothesize that since convolutional filters are inherently translation-invariant but not rotation-invariant, aligning features under different rotations reduces feature quality. Moreover, rotations are approximately 30% slower due to interpolation overhead.

| Att. Strength ($\epsilon$) | 8/255 | | 4/255 | | 2/255 | | 1/255 | |
|---|---|---|---|---|---|---|---|---|
| Metric | Top-1 | Top-5 | Top-1 | Top-5 | Top-1 | Top-5 | Top-1 | Top-5 |
| No Defense | 8.51 | 26.19 | 9.45 | 27.92 | 11.10 | 30.63 | 15.21 | 36.76 |
| Ours ($d=1$) | 18.40 | 40.36 | 19.54 | 42.65 | 21.02 | 43.86 | 24.66 | 47.81 |
| w/ Rotation | 20.01 | 41.32 | 20.84 | 42.39 | 22.10 | 43.89 | 24.72 | 46.89 |
| Ours ($d=2$) | 20.01 | 42.22 | 21.02 | 43.45 | 22.44 | 45.32 | 25.80 | 48.97 |
| w/ Rotation | 18.86 | 38.67 | 19.55 | 39.5 | 20.36 | 40.48 | 22.01 | 42.50 |
| Ours ($d=3$) | **21.17** | **43.58** | **22.04** | **44.78** | **23.44** | **46.53** | **26.58** | **49.73** |
| w/ Rotation | 15.15 | 32.89 | 15.61 | 33.46 | 16.17 | 34.19 | 17.33 | 35.35 |

Table 4: **Stochastic resonance using translation v.s. rotation on ResNet-50.** While rotations provide similar gains at low resonance levels, performance degrades as the resonance level increases, likely due to the lack of rotational invariance in convolutional filters.

Since adversarial perturbations often manifest as high-frequency noise, having a low-pass filter in early layers may form an effective defense. As our method applies to arbitrarily chosen layers, we vary the termination layer of stochastic resonance. As shown in Tab. 3, applying it only through the first residual block already achieves strong adversarial robustness, while extending it to the second block yields the strongest result. This finding is significant: running stochastic resonance through shallow layers can be sufficient as a defense strategy, which reduces overall computational cost.

**CIFAR-10-C**. Although focused on adversarial perturbations, we are also interested in understanding how stochastic resonance behaves under real-world corruptions. To this end, we evaluate our method on the CIFAR-10-C benchmark Hendrycks & Dietterich (2019), which contains a diverse set of test-time corruptions covering noise, blur, weather effects, and photometric changes. We test on a ResNet-50 model trained on the original CIFAR-10 dataset without corruption. Following the best-performing configuration identified in prior sections, with stochastic resonance applied at translation radius $d = 3$, and ensemble the features after the Residual Block 2. The dataset provides corruption severity levels from 1 to 5; we report aggregated results across all levels.

Overall, stochastic resonance recovers an average of 20.8% of the performance lost to these perturbations. In particular, our approach achieves substantial gains under several challenging corruptions, recovering 32.6% under `defocus_blur`, 34.4% under `fog`, 31.6% under `gaussian_blur`, and 31.9% under `saturate`. Other corruption types also see improvements ranging from 3.4% to 20.8%. These findings further highlight a key strength of stochastic resonance: it improves robustness across a *wide spectrum* of corruptions that are not specified during training. This property strengthens stochastic resonance as a *general-purpose*, *plug-and-play* test-time defense. Rather than being tailored to a specific adversarial formulation, it has stronger potential for real-world deployment where the exact perturbation type is often unknown and unpredictable.

| Attack Strength ($\epsilon$) | 0.02 | | | 0.01 | | | 0.005 | | | 0.002 | | |
|---|---|---|---|---|---|---|---|---|---|---|---|---|
| Metric | MAE | RMSE | D1-err | MAE | RMSE | D1-err | MAE | RMSE | D1-err | MAE | RMSE | D1-err |
| **FGSM** No Defense | 14.83 | 24.10 | 97.33 | 8.49 | 14.53 | 90.49 | 5.05 | 7.70 | 74.71 | 3.01 | 3.49 | 38.12 |
| Latent Smoothing | 13.42 | 22.09 | 96.61 | 8.12 | 13.69 | 89.25 | 4.89 | 7.05 | 73.01 | 2.89 | 3.32 | 36.25 |
| Ours ($d=1$) | 10.12 | 15.81 | 95.80 | 6.46 | 9.92 | 86.30 | 4.39 | 5.93 | 68.55 | 2.78 | 3.18 | 31.99 |
| Error Reduced (%) | 31.76 | 34.40 | 1.57 | 23.91 | 31.73 | 4.63 | 13.07 | 22.99 | 7.58 | 7.64 | 8.88 | 16.08 |
| Ours ($d=2$) | **9.22** | **13.88** | **94.61** | **6.13** | **8.87** | **84.49** | **4.19** | **5.43** | **66.74** | **2.73** | **3.12** | **31.18** |
| Error Reduced (%) | 37.82 | 42.40 | 2.79 | 27.78 | 38.95 | 6.63 | 16.92 | 29.41 | 10.66 | 9.21 | 10.65 | 18.21 |
| **PGD** No Defense | 161.70 | 162.61 | 99.99 | 131.66 | 140.55 | 98.64 | 63.97 | 88.55 | 85.31 | 6.83 | 17.03 | 39.24 |
| Latent Smoothing | 161.79 | 162.69 | 99.99 | 131.46 | 140.41 | 98.57 | 63.86 | 88.21 | 85.39 | 7.28 | 17.44 | 39.81 |
| Ours ($d=1$) | 107.86 | 125.14 | 98.29 | 69.97 | 84.72 | 91.79 | 20.66 | 42.29 | 73.21 | 4.17 | 8.32 | 29.09 |
| Error Reduced (%) | 33.30 | 23.04 | 1.70 | 46.86 | 39.72 | 6.94 | 67.70 | 52.24 | 14.18 | 38.95 | 51.15 | 25.87 |
| Ours ($d=2$) | **77.59** | **100.14** | **96.14** | **44.77** | **66.23** | **89.24** | **17.98** | **32.80** | **71.09** | **3.76** | **6.73** | **28.44** |
| Error Reduced (%) | 52.01 | 38.41 | 3.85 | 66.02 | 52.87 | 9.53 | 71.89 | 62.94 | 16.66 | 44.99 | 60.52 | 27.53 |

Table 5: **Stochastic Resonance Enhances Stereo Matching Robustness.** *Incorporating stochastic resonance significantly reduces prediction errors induced by adversarial attacks across all evaluation metrics, reducing error by up to 71.89% (MAE, when attacked by PGD with $\epsilon = 0.005$). Notably, this defense mechanism operates entirely at test time without requiring any model re-training, which sets it apart from existing methods.*

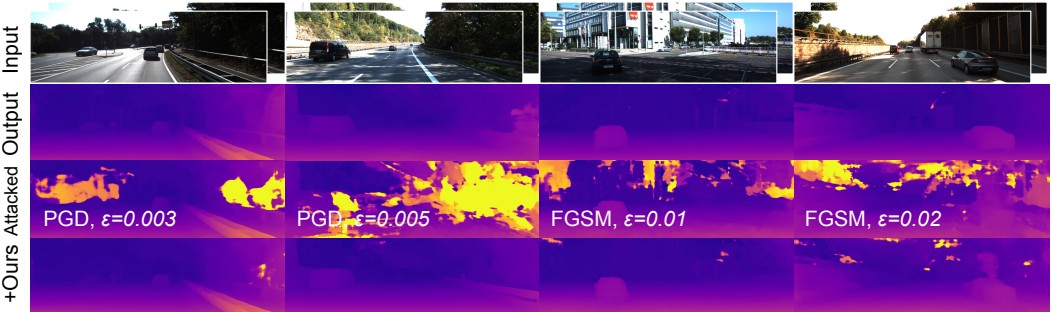

Figure 3: **Stereo matching robustness via stochastic resonance.** *We present visual results on stereo matching under various adversarial attack scenarios, including PGD and FGSM at different perturbation levels. These attacks significantly degrade the network's predictions, leading to substantial errors. By incorporating stochastic resonance, we demonstrate a significant reduction in prediction errors. This technique holds significant potential for improving robustness in safety-critical real-world applications, such as autonomous driving, where stereo vision must remain reliable under diverse environmental conditions and adversarial threats.*

## 4.2 STEREO MATCHING

Stereopagnosia Wong et al. (2021) first introduced adversarial attacks to stereo matching , yet no test-time method has demonstrated an effective defense. The primary challenge arises from the infeasibility of data augmentations, as they risk altering the physics of the input, leading to incorrect estimation. In contrast, our feature-level ensemble is suitable for this task, as transformations introduced by stochastic resonance are "undone" in the latent space, ensuring that features remain aligned precisely with the original input. This process mirrors AugUndo Wu et al. (2024) conceptually.

We evaluate our method on the standard benchmark used in *Stereopagnosia*, derived from KITTI Geiger et al. (2012). Experiments are conducted using FGSM and PGD attacks against a pre-trained PSMNet Chang & Chen (2018). Since no existing test-time defense is available, we adopt latent-space smoothing as a baseline. As shown in Tab. 5, both attacks corrupt network predictions, with PGD proving substantially more effective due to its iterative nature. Nevertheless, stochastic resonance consistently improves robustness across attack strengths. In particular, under PGD with $\epsilon = 0.005$, our method reduces errors by up to 71.89% in terms of MAE. Crucially, these gains are achieved entirely at test time, without additional training or prior knowledge of the attack.

Fig. 3 provides qualitative results, featuring different attack strengths and methods. The adversarial perturbations introduce significant distortions, as indicated by bright regions in the visualized predictions. Stochastic resonance effectively mitigates these distortions, drastically reducing prediction errors. This experiment is particularly relevant for safety-critical applications such as autonomous driving, where adversarial disturbances can arise not only from malicious attacks but also from environmental factors such as adverse weather conditions, varying illumination, or sensor degradation.

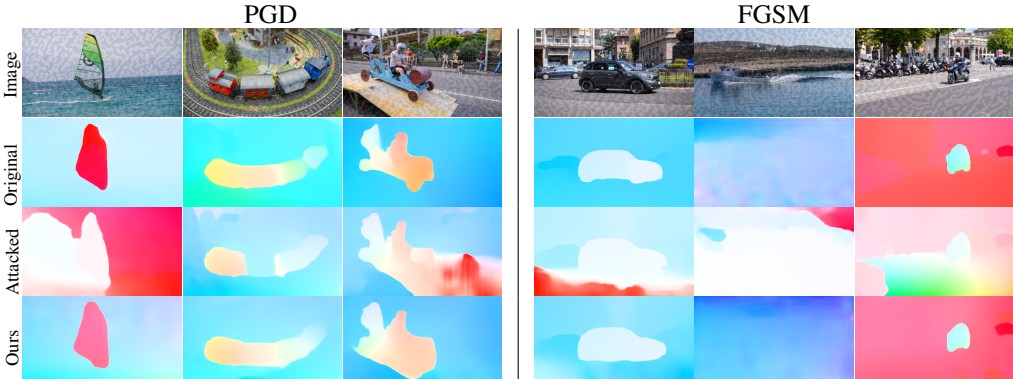

Figure 5: **Optical flow robustness via stochastic resonance.** *Qualitative results (visualized with a color wheel) show that our method substantially mitigates the degradation caused by both PGD and FGSM attacks. This robustness is particularly relevant for visual perception systems that rely on accurate motion estimation.*

While some defenses Zhang et al. (2023); Cheng et al. (2021; 2022); Berger et al. (2022) have been proposed to train a more robust network under adverse conditions, test-time defenses remain largely unexplored. Our method is the first to provide a viable solution in this setting.

### 4.3 OPTICAL FLOW

We further evaluate our method on optical flow, which computes the dense motion field between two images. The accuracy of optical flow is measured using the End-Point Error (EPE). Although several adversarial attacks on optical flow have been proposed Schrodi et al. (2022); Ranjan et al. (2019); Schmalfuss et al. (2022); Agnihotri et al. (2023); Scheurer et al. (2024), there is currently no standardized benchmark for the evaluation of defense. Therefore, we adopt a simple and controlled setup: we employ the RAFT Teed & Deng (2020) model and focus on global adversarial perturbations generated by PGD and FGSM. We test on the DAVIS Pont-Tuset et al. (2017) dataset.

To defend against adversarial attacks, we apply stochastic resonance to the convolutional feature extractor of RAFT. Since the perturbation is applied only at the feature extraction stage, no additional overhead is introduced in the computationally intensive matching module. Quantitative results (Fig. 4) show that increasing Stochastic Resonance reduces EPE, which aligns with our findings in classification. As in Fig. 5, our approach effectively removes errors caused by adversarial noise.

We further compare our method to an alternative ensembling approach that aggregates predictions in the output space, conceptually similar to TTE Pérez et al. (2021). In this variant, we apply the same stochastic transformations but instead ensemble at the output level rather than in the feature space. While this method provides marginal improvements, it remains less effective than our approach. This finding highlights the advantage of having the freedom to choose from any stage of the model to perform ensemble. In this particular experiment, we demonstrate that ensembling solely at the image encoding sub-module, while leaving the rest of the RAFT network unchanged, yields substantial improvements in robustness, thanks to the flexibility of our method.

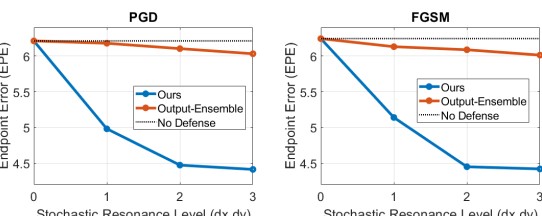

Figure 4: **Enhanced optical flow robustness with stochastic resonance.** *Under PGD and FGSM, stochastic resonance significantly reduces endpoint error in optical flow estimation. Notably, our method performs ensembling in the latent feature space rather than the output space, providing greater flexibility. While ensembling in the output space offers minor performance gains, our approach consistently achieves superior robustness across all levels of stochastic resonance.*

**Adverse weather corruptions.** For real-world deployment, we further evaluate stochastic resonance under adverse weather conditions Schmalfuss et al. (2023). This setting poses two unique challenges: (i) the corruptions are *not* optimized for the model and instead reflect natural image statistics, and (ii) the perturbations are highly localized (e.g., rain streaks, snow particles), violating the assumptions underlying our theoretical analysis that rely on small, spatially uncorrelated

perturbations. Despite these challenges, stochastic resonance still yields measurable improvements: recovering 7.5% of the accuracy loss for `snow` and 3.7% for `rain`. While, as expected, the gains are smaller than those observed under adversarial attacks, they nonetheless demonstrate that stochastic resonance provides meaningful robustness against real-world perturbations, which is consistent with our findings on CIFAR-10-C. This further confirms stochastic resonance's effectiveness as a generic test-time robustness mechanism that extends beyond adversarial settings.

## 5 DISCUSSION AND CONCLUSION

**Speed.** Stochastic resonance incurs low computational overhead when executed in parallel: raising $d$ to 3 with ResNet-50 adds only 0.06 seconds to the inference time on an NVIDIA 1080Ti GPU. Notably, most of the overhead arises from transformations (and their inverse) implemented by Python loops with `torchvision.transforms`. We expect this overhead to be drastically reduced with more efficient CUDA implementations. Excluding this `for` loop, stochastic-resonance-specific forward computation is lightweight: 0.028s with $d=3$ (49 forward passes), while a single forward pass ($d=1$) takes 0.007s. Moreover, strong robustness can be attained by applying stochastic resonance only to shallow layers, offering substantially greater efficiency than existing ensemble-based defenses (e.g. Pérez et al. (2021)) that require multiple passes through the entire network. Moreover, stochastic resonance is fully plug-and-play. In contrast, adversarial training is attack-specific, over 6× slower than standard training, and must be repeated for each type of threat. As such, the computation of our method is well justified by its robustness gains and training-free nature.

**Memory.** Increasing $d$ from 0 to 3 raises the number of forward passes from 1 to 49 but increases peak memory only moderately (from 158MB to 650MB) for ImageNet-sized inputs on ResNet-50. Modern GPUs easily accommodate this overhead. An interesting observation emerges when performing *adaptive* (worst-case) attacks: memory usage jumps from 346MB to 10.6GB for a single 224×224 image. This increase makes adaptive attacks practically infeasible in standard hardware settings, providing an inherent defensive advantage. Overall, stochastic resonance offers favorable cost–robustness trade-offs with minimal memory growth and low computational footprint.

**On-demand scaling.** One of the key strengths of our approach is its flexibility: providing a trade-off between robustness and computational cost. We offer a tunable "knob" that allows practitioners to adjust the level of resilience based on available resources on the fly: when the system has more computational capacity, add a higher level of stochastic resonance, vice versa. Note that, such a design does not *rely* on additional computation, yet more computation can bring *extra* performance. Moreover, our experiments show that the method generalizes across a wide range of tasks and architectures that include an encoder. This on-demand scaling mirrors inference-time scaling in language models, where performance can be improved without modifying the underlying pre-trained model.

**Limitations.** Despite its strengths, our method has some limitations. First, while we offer parallel computation as a remedy, the computational overhead introduced by stochastic resonance may not be negligible for scenarios with memory and power constraints. Also, our current study focuses on integer-pixel translations. While this choice avoids interpolation artifacts and preserves spatial consistency, more generic transformations, including learned transformations, could be explored.

**Conclusion.** In this work, we present a signal-processing perspective for defending against adversarial attacks, motivated by the connection between adversarial perturbations and aliasing artifacts. Accordingly, we propose a "combat noise with noise" approach by introducing stochastic resonance as a defense mechanism. We formalize the problem and implement stochastic resonance using pixel-level translations paired with their inverse transformation in the feature space. The resulting method is training-free, agnostic to both tasks and attack types, and independent of network architectures.

We evaluate our method across various tasks. Empirical results on image classification demonstrate that our stochastic resonance approach achieves state-of-the-art robustness against diverse attack types, offering a clear advantage over feature-level denoising and filtering. Even in the adaptive adversary scenario, where an attacker is aware of the use of stochastic resonance, our method maintains strong robustness. Furthermore, we are the first to introduce test-time defense to dense prediction tasks. Specifically, we apply this method to stereo matching and optical flow, achieving up to a 71% reduction in prediction error. More importantly, these findings highlight the practical potential of stochastic resonance as a universal defense in real-world adversarial scenarios.

## REPRODUCIBILITY STATEMENT

We provide sufficient technical details in the paper to ensure reproducibility. Specifically, we describe the augmentations used for stochastic resonance, including augmentations (e.g. translation, rotation) and their corresponding inverse transformations, as well as the model architectures, datasets, and the network layers where our method is applied. Attack settings and evaluation protocols are drawn directly from standard benchmark datasets and publicly available code base, ensuring comparability with prior work. All implementation details necessary to reproduce our experiments, including parameters and ablation settings, are provided in the main paper and further expanded in the Appendix. Our experiments can be reproduced on a single desktop-level GPU without requiring large-scale computational resources. We will release the complete source code and pre-computed adversarial data upon publication.

## LLM STATEMENT

All technical content of this work, including literature review, methodology, experiments, and analyses, was developed entirely by the authors. Large Language Models (LLMs) were employed as a tool for proofreading, without contributing to the scientific or technical substance of the manuscript.

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

## A  IMPLEMENTATION FOR CLASSIFICATION

For our classification experiments, we built our implementation on top of standard network architectures, implementing SR on two main architectures derived from the FSR codebase Kim et al. (2023):

- ResNet-18 He et al. (2016): A standard residual network with 18 layers organized in four main blocks with increasing channel dimensions (64, 128, 256, 512).
- WideResNet-34 Zagoruyko & Komodakis (2016): A wider variant of ResNet with depth 34 and width factor 10, resulting in higher representational capacity with three main blocks with channel dimensions scaled by the width factor (160, 320, 640).

For both architectures, we apply SR at the bottleneck layer (after the final convolutional block).

Given an input image batch $x \in \mathbb{R}^{b \times 3 \times h \times w}$ (where $b$ is the batch size), our SR approach operates as follows. First, we create a set of $(2d_x + 1) \times (2d_y + 1)$ perturbed versions of the input by applying pixel-level translations within the range $[-d_x, d_x] \times [-d_y, d_y]$ pixels:

$$X_{\text{perturbed}} = \{g_{i,j}(X) \mid i \in [-d_y, d_y], j \in [-d_x, d_x]\} \tag{2}$$

where $g_{i,j}$ translates the image by $(i, j)$ pixels. These transformations are applied using PyTorch's "transforms.functional.affine" function with translation parameters while preserving the original image properties.

All images are concatenated into one batch and processed through the network in parallel up to the bottleneck layer:

$$F = \phi(X_{\text{perturbed}}) \tag{3}$$

where $\phi$ represents the network up to the bottleneck layer. This batch processing approach significantly improves computational efficiency compared to individual forward passes.

After obtaining feature maps for all perturbed inputs, we aggregate them to create a single enhanced feature map:

$$F_{\text{ensembled}} = \frac{1}{n} \sum_{i,j} T_{-i,-j}(F_{i,j}) \tag{4}$$

where $T_{-i,-j}$ represents the inverse spatial shift operation that realigns the feature map and $n$ the number of augmentations.

Our implementation requires $(2d_x + 1) \times (2d_y + 1)$ forward passes through the network up to the bottleneck layer.

For evaluation, we tested our approach against standard adversarial attacks (FGSM Goodfellow et al. (2014), PGD-20 and PGD-100 Madry et al. (2017), and C&W Carlini & Wagner (2017)), all bounded within $\epsilon = 8/255$ under $\ell_\infty$-norm. We also report an Ensemble metric that measures the worst-case performance across all attacks for each test example, providing a comprehensive robustness assessment.

## B  IMPLEMENTATION FOR STEREO MATCHING

For our stereo matching experiments, we built our implementation on top of standard stereo network architectures to ensure our approach remains model-agnostic and requires no training or fine-tuning. We integrated SR with PSMNet Chang & Chen (2018), a pyramid stereo matching network with a stacked hourglass architecture that uses 3D convolutions on a cost volume constructed by concatenating features. We apply SR at the feature extraction stage, before cost volume construction, where stereo correspondences are first established.

Given a pair of input stereo images $x_L, x_R \in \mathbb{R}^{b \times 3 \times h \times w}$ (where $b$ is the batch size), our SR approach operates as follows. First, we create a set of $(2d_x + 1) \times (2d_y + 1)$ perturbed versions of each input image by applying translations within the range $[-d_x, d_x] \times [-d_y, d_y]$ pixels:

$$X_{L,\text{perturbed}} = \{g_{i,j}(x_L) \mid i \in [-d_y, d_y], j \in [-d_x, d_x]\} \tag{5}$$

$$X_{R,\text{perturbed}} = \{g_{i,j}(x_R) \mid i \in [-d_y, d_y], j \in [-d_x, d_x]\}. \tag{6}$$

All images are concatenated into batches and processed through the feature extraction component of the network:

$$F_L = \phi(X_{L,\text{perturbed}}) \tag{7}$$
$$F_R = \phi(X_{R,\text{perturbed}}) \tag{8}$$

where $\phi$ represents the feature extraction component of the stereo network. This batch processing approach significantly improves computational efficiency compared to individual forward passes.

After obtaining feature maps for all perturbed inputs, we aggregate them to create a single enhanced feature map.

Our implementation requires $(2d_x + 1) \times (2d_y + 1)$ forward passes through the feature extraction component of the network for each stereo image.

For evaluation, we tested our approach against adversarial attacks generated using FGSM Goodfellow et al. (2014) and I-FGSM Kurakin et al. (2018) (a special case of PGD), bounded within various $\epsilon$ values ($\{0.002, 0.005, 0.01, 0.02\}$) under $\ell_\infty$-norm. We measured performance using three standard stereo matching metrics: Mean Absolute Error (MAE), Root Mean Square Error (RMSE), and D1-error (percentage of pixels with disparity error greater than 3 pixels or 5% of the ground truth) Luo et al. (2018).

## C    DETAILS ABOUT ATTACK ON OPTICAL FLOW

To find an adversarial attack for optical flow estimated by a given neural network $f$, we utilize a similar approach to Oskouie et al. (2024) that aims to find a perturbation $\delta$ for given frames $F_1$ and $F_2$, maximizing the discrepancy between predicted and ground-truth optical flow $OF$. If the ground-truth optical flow is unavailable, we use the predicted optical flow from the unattacked frame as our surrogate ground-truth. Our method applies $\delta$ to the first input frame, then uses a deep neural network to estimate optical flow from the perturbed frames. The objective is to maximize the average end-point error (EPE) between the predicted and ground-truth optical flow, calculated as the mean Euclidean distance between corresponding 2D flow vectors. In other words, the $\epsilon$-norm bounded adversary $\delta$ for optical-flow is calculated by optimizing the following

$$\max_{\delta:\|\delta\| \leq \epsilon} \text{EPE}\big(OF, f(F_1 + \delta, F_2)\big). \tag{9}$$

One $l_\infty$-bounded adversary $A$ for the aforementioned optimization problem is Fast Gradient Sign Method (FGSM) Goodfellow et al. (2014) which can be obtained by

$$\text{L} = \text{EPE}(OF, f(F_1, F_2)),$$
$$A = F_1 + \epsilon \cdot \text{sign}\big(\nabla_{F_1}\text{L}\big). \tag{10}$$

Projected gradient descent (PGD) Madry et al. (2017) represents an enhanced and more complex version of FGSM. This attack method generates adversarial examples through an iterative process and the formulation for this attack is as following

$$F_1^{(t+1)} = \Pi_{F_1+\mathcal{S}}\big(F_1^{(t)} + \alpha \cdot \text{sign}(\nabla_{F_1}\text{L})\big). \tag{11}$$

Note that in PGD, since the perturbations are considered to be too minimal to significantly alter the flow dynamics, the ground-truth optical flow is not updated by intermediate perturbations applied to the input data.

For our experimental setup, we chose to set the norm value $\epsilon$ at $\frac{10}{256}$. Furthermore, we configured the PGD algorithm to run for 10 iterations. The step size $\alpha$ was determined by dividing $2.5 \cdot \epsilon$ by the total number of iterations, ensuring a balanced progression throughout the optimization process.

## D    ADDITIONAL VISUALIZATIONS

Here we provide additional visualizations from our experiments comparing SR under both FGSM and PGD attack. We also provide a visual showing the results of FGSM and PGD pertubation on various images.

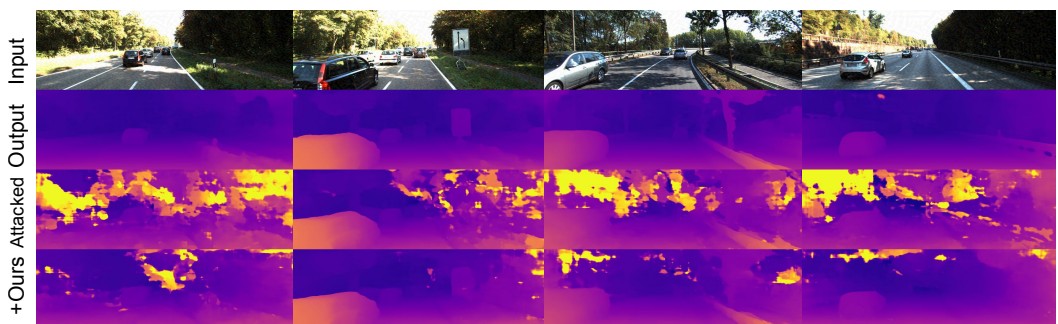

Figure 6: Visual results on stereo matching against FGSM attack, without and with SR, $\epsilon = 0.02$

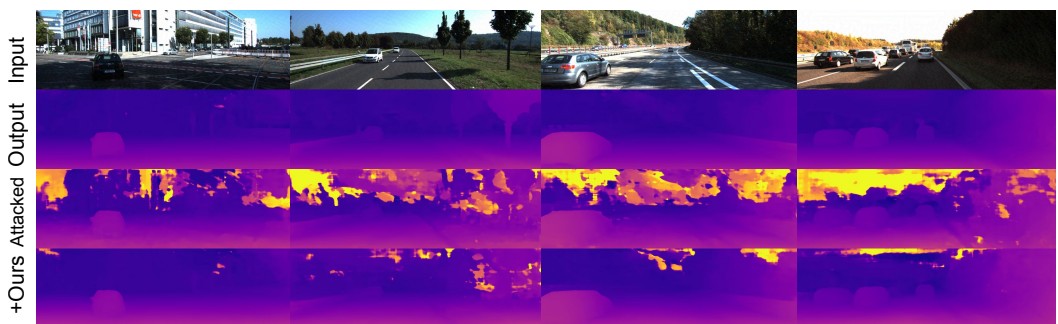

Figure 7: Visual results on stereo matching against FGSM attack, without and with SR, $\epsilon = 0.01$

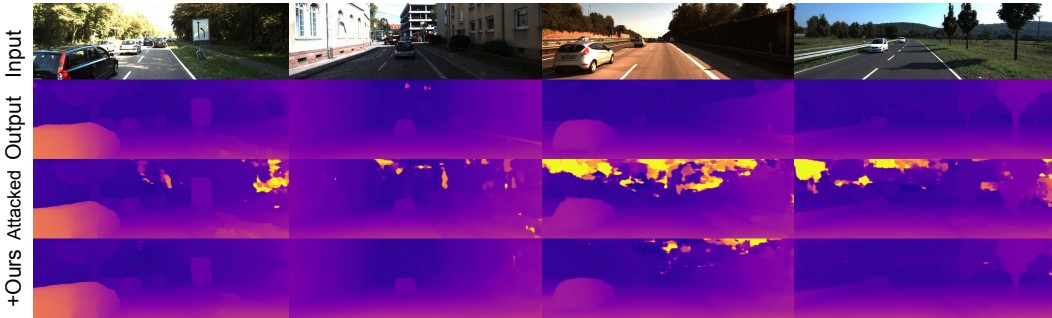

Figure 8: Visual results on stereo matching against FGSM attack, without and with SR, $\epsilon = 0.005$

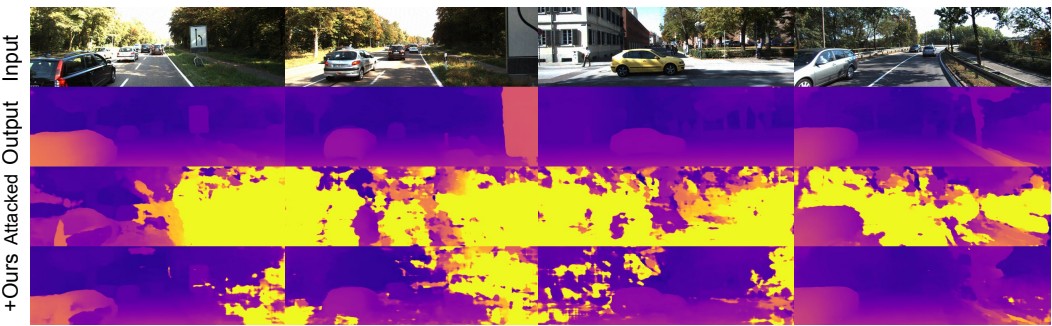

Figure 9: Visual results on stereo matching against PGD attack, without and with SR, $\epsilon = 0.01$

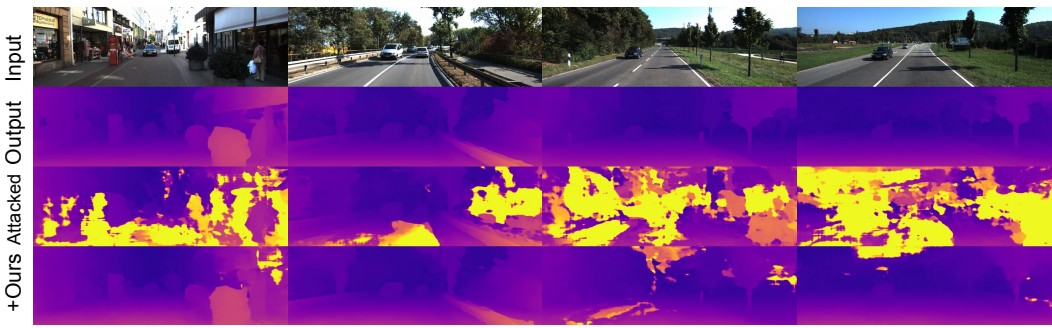

Figure 10: Visual results on stereo matching against PGD attack, without and with SR, $\epsilon = 0.005$

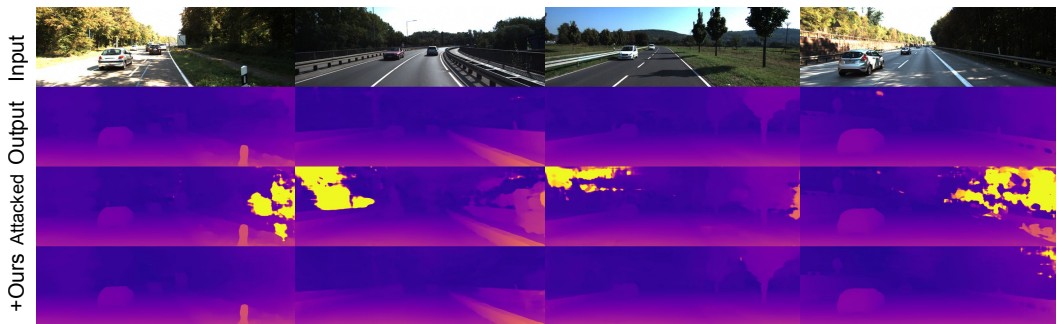

Figure 11: Visual results on stereo matching against PGD attack, without and with SR, $\epsilon = 0.002$

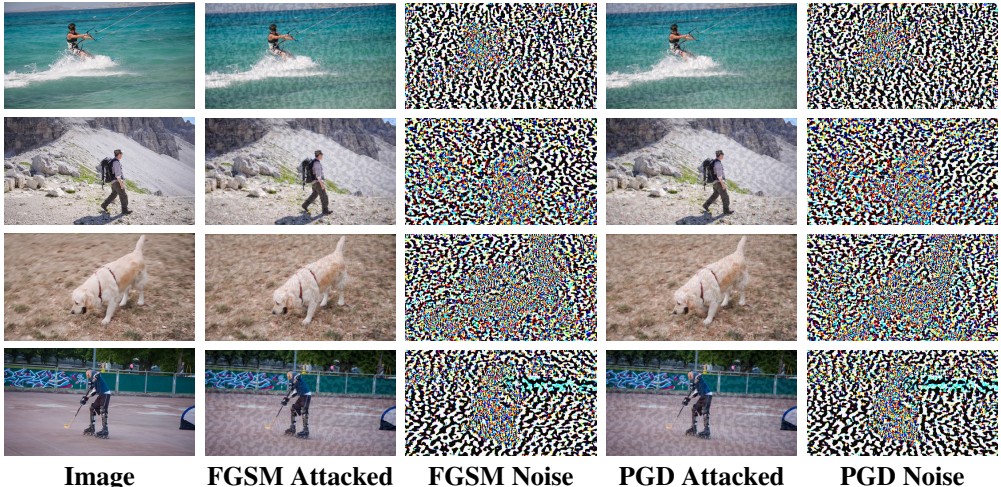

Figure 12: Original images and their corresponding attacked images and perturbations using FGSM and PGD methods on optical flow. The attacks mostly target the main object observed in the image.

