# OpenReview forum: "Test-Time Defense Against Adversarial Attacks via Stochastic Resonance of Latent Ensembles"
_ICLR.cc/2026/Conference — Submitted to ICLR 2026_

### Official Review · Reviewer_HimK · 2025-10-20

**Soundness:** 2
**Presentation:** 2
**Contribution:** 3
**Rating:** 4
**Confidence:** 4

**Summary:**

This paper proposes a training-free test-time defense mechanism based on stochastic resonance (SR) in latent feature space. The method perturbs input images with small translations, aligns the resulting embeddings, and aggregates them to improve robustness without retraining or architectural changes. Experiments on classification (CIFAR-10, ImageNet) and dense prediction tasks (stereo matching, optical flow) are provided.

**Strengths:**

1. The idea of “combat noise with noise” via stochastic resonance is conceptually novel and elegantly simple.

2. The framework is easy to integrate into existing architectures and does not require retraining.

3. The paper provides extensive experimental results across multiple tasks, including dense prediction, which is less explored in adversarial defense.

These strengths collectively highlight the method’s potential practical value as a lightweight and versatile test-time defense strategy.

**Weaknesses:**

1. This paper lacks a rigorous theoretical foundation to support the claimed robustness improvement. While the intuition of “combating noise with noise” is interesting, no analytical explanation or formal proof is showing why the proposed stochastic resonance mechanism effectively suppresses adversarial perturbations.

2. Although this method is described as training-free, architecture-agnostic, and attack-agnostic, the experimental scope is confined to same-dataset evaluations. The experiments does not include critical transfer-setting tests.

**Questions:**

1. See in W1. Could the authors provide at least a theoretical analysis?

2. See in W2. How would the proposed approach perform under realistic transfer settings, such as cross-dataset, cross-resolution, or cross-style experiments?

3. The main schematic (Figure 1) is not referenced in the method section.

4. Clarify whether the claimed robustness persists against unseen attack families.

The motivation of the article is quite reasonable, if all of my concern are addressed, I will increase my score.

**Details Of Ethics Concerns:**

No.

---

> ### Author Response · Authors · 2025-11-21
> **Thank you for your constructive review!**
>
> **Q1**: Please see official comments to all reviewers for a theoretical analysis of the effectiveness of Stochastic Resonance
>
> **Q2**: Thank you for the suggestion. As recommended by the reviewer, we evaluated our method on **CIFAR-10-C** (also suggested by R3), which contains a diverse set of real-world corruptions covering noise, blur, weather effects, and photometric changes. Across corruption types and severity levels, SR recovers an average of **20.8%** of the performance lost to these perturbations.
>
> In particular, SR shows strong improvements under:
> - **defocus blur:** +32.6% recovered
> - **fog:** +34.4% recovered
> - **gaussian blur:** +31.6% recovered
> - **saturate:** +31.9% recovered
>
> Other corruption types also benefit, with gains ranging from **3.4% to 20.8%**. These results demonstrate that SR is effective not only against adversarial attacks, but also against a wide spectrum of realistic corruptions during the imaging process, addressing cross-dataset, cross-style, and cross-perturbation generalization.
>
> We additionally evaluated SR on optical flow under adverse weather conditions (snow and rain) on the **Sintel** dataset. Under these settings, SR recovers **7.5%** of accuracy loss for snow and **3.7%** for rain on the DistractingDownpour corruption (suggested by R2).
>
> If there are specific methods or transfer settings the reviewer would like us to include, we would be happy to add them.
>
> **Q3**: We will revise.
>
> **Q4**: Our method does not require any knowledge of the attack family and is therefore always evaluated on **unseen attacks**. In the main paper, we demonstrate robustness across multiple adversarial families, including PGD, FGSM, C\&W-style losses, and spatial attacks. To further validate this point, as shown in **Q2**, we additionally include results on CIFAR10-C, and observe the same consistent robustness improvements. The perturbations in CIFAR10-C span a wide range of unseen corruption types, including `brightness`, `contrast`, `defocus_blur`, `elastic_transform`, `fog`, `frost`, `gaussian_blur`, `gaussian_noise`, `glass_blur`, `impulse_noise`, `jpeg_compression`, `motion_blur`, `pixelate`, `saturate`, `shot_noise`, `snow`, `spatter`, `speckle_noise`, and `zoom_blur`. We are happy to incorporate these results into the revised manuscript.
>
> We hope the expanded theoretical analysis and new experiments address the reviewer’s concerns. We are happy to continue the discussion!

---

> > ### Comment · Reviewer_HimK · 2025-11-25
> >
> > Thank you for the response. However, my original question was specifically about transferability under distribution shift, including cross-dataset, cross-resolution, and cross-style settings. CIFAR-10-C is useful, but it does not fully represent these forms of transfer. So I will maintain my score.

---

> ### Author Response · Authors · 2025-11-25
>
> There might be a misunderstanding. We do already test on cross-dataset, cross-style, and cross-resolution cases. To clarify our current coverage:
>
> **Cross-dataset.**
> Our optical flow evaluation already follows the cross-dataset benchmark in the community: training on FlyingThings and testing on DAVIS (sim-to-real).
>
> **Cross-style.**
> CIFAR-10-C includes 19 heterogeneous distribution shifts spanning styles, weather, blur, noise distributions, and photometric changes. These are unseen at training time, and constitute far larger distribution shifts than those typically evaluated in adversarial attacks. This directly addresses transferability to unseen data domains and perturbation processes.
>
> **Cross-resolution.**
> Our optical flow is cross-resolution by default. The training set has a resolution of 768 by 368, and we test on 432 by 240. We are not aware of any specific established cross-resolution evaluation standard for adversarial defenses. Without a reference, we cannot determine what the reviewer considers the "correct" setting.
>
> If this is not the cases that the reviewer is looking for, then we are happy to conduct further experiments, but we will need further guidance from the reviewer. After doing a deep drive through the literature, we were unable to identify prior work that uses the forms of transfer the reviewer mentioned (cross-dataset, cross-resolution, cross-style) in the adversarial robustness literature, especially for test-time defenses. Without a clear protocol or reference, it is not possible to know what specific experiments the reviewer expects.

---

> > ### Comment · Reviewer_HimK · 2025-11-28
> >
> > Thank you very much for the detailed clarification. To avoid misunderstanding, I clarify what I meant by cross-style. I was referring to style-level domain shifts across visually distinct modalities, such as sketch to real, painting to photo, which is similar to the variations present in DomainNet or other multi-style domain adaptation benchmarks. If the authors can include these experiments, that would strengthen the paper.

---

> > > ### Author Response · Authors · 2025-12-01
> > > **Additional results on domain adaptation**
> > >
> > > We appreciate the clarification. Respectfully, we emphasize that our paper is on adversarial robustness and test-time defenses, not domain adaptation. While stochastic resonance is a generic mechanism that can in principle be applied broadly, evaluating every possible distribution-shift scenario is beyond the intended scope of the paper and was never claimed as part of our technical contributions.
> > >
> > > That said, to address the reviewer's concerns, we conducted an additional **sim-to-real, cross-resolution** evaluation. Specifically, we applied stochastic resonance to **DepthAnything V2**, trained on the synthetic **VKITTI** dataset, and evaluated on real-world **KITTI** depth prediction. This constitutes both (1) a cross-style transfer (sim to real) and (2) a cross-resolution transfer (KITTI images downsampled by 0.5), aligning with the reviewer's request.
> > >
> > > The quantitative results below show that stochastic resonance consistently improves performance across all standard depth metrics:
> > >
> > > | **Method**        | **AbsRel ↓** | **SqRel ↓** | **RMSE ↓** | **RMSE-log ↓** | **δ1 ↑** | **δ2 ↑** | **δ3 ↑** |
> > > |-------------------|--------------|-------------|-------------|----------------|----------|----------|----------|
> > > | original          | 0.2376       | 1.3418      | 5.3753      | 0.2420         | 0.627    | 0.950    | 0.991    |
> > > | +ours (d = 1)     | 0.2231       | 1.2769      | 5.3274      | 0.2325         | 0.669    | 0.953    | 0.991    |
> > > | +ours (d = 2)     | 0.2118       | 1.2217      | 5.2746      | 0.2252         | 0.700    | 0.955    | 0.991    |
> > > | +ours (d = 3)     | **0.2028**   | **1.1789**  | **5.2296**  | **0.2196**     | **0.721**| **0.957**| 0.991|
> > >
> > > Notably, the best setting (d = 3) yields a **14.6%** reduction in AbsRel relative to the original model, along with consistent improvements across all other metrics. These results demonstrate that our approach remains effective under dataset, style, and resolution shifts.

---

### Official Review · Reviewer_2PRW · 2025-10-29

**Soundness:** 2
**Presentation:** 2
**Contribution:** 3
**Rating:** 4
**Confidence:** 3

**Summary:**

This paper uses the concept of stochastic resonance (SR) from signal processing as a test-time defense strategy. The attacked image is translated on an integer-pixel basis and combined with SR by encoding the translated images, upsampling, and then inversely translating them. These results are then aggregated for the downstream trask prediction. This approach is applied to trained networks during inference time, with the focus on applying this technique on already adversarially trained networks. Experiments conducted on CIFAR-10 using common adversarial attacks, such as PGD-20/100, in combination with adversarially trained ResNet variants, demonstrate improved defense abilities. Additionally, this method is also used for stereo matching and optical flow defenses.

**Strengths:**

- This paper applies a classical signal processing method in a novel setting for test-time defense
- Various settings and examples including ablation about the level of translations and which network layer to use as a feature extractor are shown
- The proposed method is flexible to any (adversarially) pretrained network, making it eventually applicable to new types of networks and training schemes
- The proposed method shows advancements over existing defense techniques

**Weaknesses:**

- The experiments miss comparisons to the more widely used AutoAttack (Croce and Hein, 2020).
- Table 2 and 3 lack a comparison to other approaches that don’t use adversarial training.
- Different layer features used as the embedding lead to different results, which makes sense, but this constraint limits the easy usage of this method.
- This paper primarily focusses on CNNs. Is there a reason for this? Current methods often utilise transformer-based networks due to their inherent robustness from the start. Providing more information on how the proposed approach performs across different networks, while maintaining consistent strategies (such as the type of adversarial training), would better demonstrate the strengths and weaknesses of the proposed method.
- Ukita and Kenichi, 2023 also explore feature-space stochasticity as both an adversarial attack and a defense method. Since this paper focusses explicitly on feature-space adversarial examples, it would be beneficial to include either the adversarial attack or a comparison to this defense strategy.
- Section 3 needs improvement. It’s difficult to grasp the novelty of the paper and how SR is used. The explanation is tedious to understand.


*Smaller points*:
- Figure 2 is too small and hard to read
- The captions are in an unusual style.

*Missing literature*:
- Reliable evaluation of adversarial robustness with an ensemble of diverse parameter-free attacks, F. Croce and M. Hein,  ICML 2020
- Boosting Adversarial Robustness with CLAT: Criticality-Leveraged Adversarial Training, B. Gopal, H. Yang,  J. Zhang,  M. Horton, Y. Chen, ICML 25
- An automated robust fine-tuning framework, X. Xu, J. Zhang, M. Autolora Kankanhalli, ICLR 2024
- Adversarial attacks and  defenses using feature-space stochasticity, J. Ukita and O. Kenichi,  Neural Netw., Vol. 167, pp. 875-889, 2023

**Questions:**

- Which $L_p$ norms were used? These specifics are missing
- How many runs were conducted?
- A clarification which noise distribution is used for SR is needed- Line 260f suggests that the translations are the perturbation. So, is there any noise added during SR as in SRT, as mentioned in Lao et al. (2024)? Providing more information about this would make the approach clearer.
- How about other attacks and corruptions at test time, such as CIFAR-10-C?
- What is the optimal layer depth for this approach?
- In general I feel like the SR section and thus the method section itself could need more clarification, for easier accessing the novelty of this paper.

---

> ### Author Response · Authors · 2025-11-21
> **Thank you for your constructive review!**
>
> **W1**: Thank you for this suggestion. We have run experiments using **AutoAttack**. The results are shown below:
>
> | Method | AT | +FD | +CAS | +CIFS | +FSR | **+SR (Ours)** |
> |--------|------|------|------|--------|--------|----------------|
> | **AutoAttack Robust Acc.** | 44.11 | 44.57 | 44.23 | 43.94 | 46.41 | **49.78** |
>
> Our SR defense outperforms all baselines by a substantial margin under AutoAttack. We will add these results and the corresponding discussion to the revised manuscript.
>
> **W2**: For the settings in Tables 2 and 3, we are not aware of existing test-time baselines that report results under the same tasks and evaluation protocols. If the reviewer has specific methods they would like us to include, we would be happy to run those comparisons and incorporate them into the revised manuscript.
>
> **W3**: We are not sure of where any limitation of our method would arise, since applying it to any layer is not a constraint but added flexibility: By default, all the experiments shown, except for Table 3, are for the last-layer activations, but the user has the freedom to apply it to any other layer if so desired.
>
> **W4**: Our method is architecture-agnostic, and in principle applies to CNNs, ransformers, and hybrid pipeline. The reason most of our reported results use CNN backbones is that standard robustness benchmarks and widely adopted Tbaselines in adversarial evaluation remain CNN-based, making them the most directly comparable to prior work. As the reviewer rightfully marked in the strengths, our method is applicable to new types of networks and training schemes.
> As suggested by the reviewer, we have added an additional experiment with Vision Transformers, which also confirms our claim. Using a pre-trained DINO ViT/S encoder, we apply adversarial perturbations that maximally distort the latent embedding, and then apply SR with increasing translation radius $d$. The $L_1$ distance between adversarial and clean embeddings is summarized below:
>
> | **d**               | 0 | 1 | 2 | 3 | 4 |
> |-------------------------|-------|-------|-------|-------|-------|
> | **L1 Distance (Adv vs. Clean)** | 12.30 | 5.29 | 3.50 | 2.64 | 2.12 |
>
> This monotonic reduction is consistent with our theoretical analysis and demonstrates that SR also improves robustness for transformer-based encoders such as DINO.
>
> **W5**: We will cite and discuss Ukita & Kenichi (2023) in the revised manuscript. However, we were unable to locate publicly available code or implementation details for their method, which prevents us from running a direct empirical comparison.
>
> **W6**: We have provided additional theoretical analysis in the official comment to all reviewers that explains the core novelty of our approach and how SR is incorporated into the model. We will revise Section 3 accordingly, streamline the exposition, and integrate this expanded explanation into the updated manuscript during the discussion period.
>
> **Q1**: We use $ L_{inf} $ norm to construct the adversarial perturbations. We will clarify.
>
> **Q2 & Q3**: we use uniformly sampled translations. To avoid interpolation artifacts and isolate the effect of SR itself, we simply traverse all integer offsets in a small grid, which makes the procedure deterministic and serves as a clean proof of concept. Therefore, we do not need multiple runs. No additional pixel-space noise is added in the current experiments. However, the framework is fully compatible with continuous sampling or broader noise distributions, and stochastic sampling can be used for computational efficiency if desired.
>
> **Q4**: As recommended by the reviewer, we evaluated our method on **CIFAR-10-C**, which contains a diverse set of test-time corruptions. Across corruption types and severity levels, SR recovers an average of **20.8%** of the performance lost to these perturbations.
>
> In particular, SR shows strong improvements under:
> - **defocus blur:** +32.6% recovered
> - **fog:** +34.4% recovered
> - **gaussian blur:** +31.6% recovered
> - **saturate:** +31.9% recovered
> Other corruption types also benefit, with gains ranging from **3.4% to 20.8%**. These results indicate that SR is effective not only against adversarial perturbations, but also against a wide spectrum of real-world test-time corruptions.
>
> **Q5**: In general, we apply the method to the last layer as default. There may be applications where one wishes to alter different spatial frequencies, or scales, or level of granularity, which is domain-specific and a subjective user choice.

---

> > ### Comment · Reviewer_2PRW · 2025-11-27
> > **Official comment by reviewer 2PRW**
> >
> > I thank the authors for their answers.
> > I appreciate the adidtional experiments and the additional theoretical analysis, which adress my biggest concerns.
> > I would appreciate if these additional information could be also included in an updated version, which would convince me to update my score.

---

> > > ### Author Response · Authors · 2025-12-01
> > > **Thank you for your support.**
> > >
> > > Thank you for your support, and we are glad that the additional experiments and theoretical analysis addressed your concerns. We have provided an updated version of the paper that incorporates all new material, including the expanded theoretical explanation, CIFAR-10-C results, and AutoAttack evaluations. Although the discussion period was unexpectedly shortened, we sincerely appreciate your constructive feedback.

---

### Official Review · Reviewer_6Kns · 2025-10-29

**Soundness:** 3
**Presentation:** 3
**Contribution:** 4
**Rating:** 8
**Confidence:** 4

**Summary:**

The paper proposes a test-time defense against adversarial perturbations by latent ensembling via stochastic resonance. The defense is training-free, plug-and-play at inference, and can be easily applied to e.g., an encoder block, yielding improved robustness to multiple standard adversarial attacks. On a technical level, it averages latent embeddings of purposefully transformed inputs (small integer-pixel translations) to cancel the effect of adversarial noise. Experiments cover diverse applications and models, ranging from image classification (CIFAR-10, ImageNet; multiple backbones) and stereo matching (PSMNet) to optical flow (RAFT). The paper shows that defended method remain competitive under adaptive worst-case attacks.

**Strengths:**

**Originality:** Using stochastic resonance as purposeful perturbations to reduce the influence of extraneous adversarial noise is elegant and grounded in signal-processing intuition (aliasing vs. adversarial noise). The formalization and Eq. (1) are clear. It is a quite neat conceptual twist.

**Quality:** The quality of the experimental evaluation is high. It covers a broad range of problems, namely classification (CIFAR-10 with AT/TRADES/MART; ImageNet with ResNet-50 and ViT-Small), stereo (PSMNet), and optical flow (RAFT), though not many models per task (see weaknesses). The proposed method outperforms strong baselines and TTE, showing consistent gains over FD/CAS/CIFS/FSR and output-space TTE. The experiments also includes a worst-case analysis demonstrating that the method remains robust under these adaptive attacks, which is technical rigorous but often overlooked step when proposing new defense mechanisms.

**Clarity:** The paper is well written and easy to read, with figures and tables that support the made claims.

**Significance:** The proposal of a training-free and plug-and-play defense against pixel-level attacks, which performs well across various problems, is a significant step towards defending methods from pixel-noise. The method design makes it easy to apply across different backbones and tasks (though more details would be helpful, see weaknesses), and ensembling features in shallow layers helps reduce costs. Especially the tests on stereo and optical flow are great additions to the classification problem, and demonstrate the broad applicability of the method.

**Weaknesses:**

**Experiments**
- While a few classification models were tested, only one model is tested for the stereo and optical flow problems. To demonstrate the broad applicability, it would help to report results for more methods on those domains.
- The method and especially the group actions are introduced very generally, but most results rely on integer translations only. It would be nice to also consider broader, learned or task-symmetry-aware groups. Furthermore, for rotations, robustness degrades at higher SR levels (Table 4) and is slower due to interpolation. This undercuts the “on-demand scaling” claim as currently phrased in the discussion.
- The compute analysis is not comprehensive enough. The paper reports a delta time (+0.06 s at SR-3 on 1080Ti), but no baseline absolute inference times or throughput (img/s) across models and SR levels, relative increase in inference time, or memory footprint analysis, and no breakdown of parallelism limits on commodity GPUs. Additional statistics on compute would be helpful.

**Plug-and-Play nature of method**
- As evidence of the method’s plug-and-play nature, I would be helpful to be more specific on how to implement the method for optical flow or stereo methods, and if this implementation would be different for individual optical flow methods.

**Scope and Related Work**
- The positioning of this work vs. prior TTE could be sharper. The paper states output-space ensembling (TTE) is a special case and less effective, but an experiment for direct comparison is only done for classification on CIFAR-10. A more detailed comparison is only hinted at in lines 417 ff.
- There are a few more references, listed under Questions - minor comments, that appear relevant to the paper’s scope.

**Questions:**

- L.451 reports +0.06 s at SR-3 on ResNet-50 (1080Ti) and 0.095 s sequentially. What are the baseline inference times ? And would it be possible to report memory usage and throughput (img/s) across SR levels?
- L.458 contrasts inference-time cost with training time of adversarial training (6x longer than vanilla training). For a budget comparison that is relevant for deployment, what is the comparison to e.g., test-time TTE or feature-denoising methods at the same latency?
- Regarding the claimed “on-demand scaling” (L.460): Table 4 shows monotonic gains for translations but drops for rotations at higher SR. Is there a bound when more SR helps? Also, I would appreciate clarification on the claim that “on-demand scaling” does not extend to rotations and possibly other group transformations.
- It would be helpful to provide more details on the optical flow implementation. For the RAFT experiments, was the same integer translation applied to both frames? Where do inverse alignment and aggregation occur (pre-correlation vs post-correlation)?
- The paper argues that PGD is stronger than localized patch attacks for optical flow. Do SR gains persist for localized patch attacks on stereo and optical flow? It would be especially interesting to see whether the method works for localized attacks as well, as it is to be expected to work better with global perturbations like PGD. As localized attacks are cases where spatial ensembling might behave differently or fail, this test might lead to interesting insights.

**Minor Comments:**
- Figure 1 is not referenced inline
- Inline citations miss parenthesis
- CosPGD [Agnihotri et al., ICML’24] and DistractingDownpour [Schmalfuss et al., ICCV’23] are other established attacks for optical flow
- Static defenses have also been studied specifically for optical flow in [Scheurer et al. “Detection defenses: An empty promise against adversarial patch attacks on optical flow” WACV’24]
- The idea of countering adversarial noise with noise was also used for action recognition in [Zhang et al. “Adversarially Robust Video Perception by Seeing Motion”, Arxiv’22].

---

> ### Author Response · Authors · 2025-11-21
> **Thank you for your support! (1/2)**
>
> **W1**: In addition to our 122 tests  (3 tasks, 4 datasets, 6 architectures and variants, and 8 adversarial settings), it has suggested  we additionally test SR under **DistractingDownpour** [Schmalfuß et al., ICCV’23] and **Detection Defenses** [Scheurer et al., WACV’24]. With snow corruption, SR recovers **7.5%** of accuracy loss; with rain, **3.7%**; and with patch attacks, **5.8%** (see Q5 for details). We also plan to include optical flow comparisons, but more importantly we will share the code so anyone can test the model.
>
> **W2**: Thank you for the suggestion. As the reviewer rightfully noted, rotation-based SR improves robustness only up to a point; beyond that, interpolation artifacts begin to dominate, which explains the degradation seen in Table 4. Exploring broader, learned, or task-specific transformation groups is indeed a promising direction, and we view the design of new SR perturbations as an important avenue for future research.
>
> **W3**: As requested, we additionally present a consolidated compute and memory analysis using \textbf{ResNet-50} under different translation radii $d$, evaluated on both CIFAR-sized inputs (32$\times$32) and ImageNet-sized inputs (224$\times$224). All measurements are obtained using `torch.cuda.max_memory_allocated(device)` to capture peak memory footprint. This illustrates how computational budget scales with SR depth.
>
> | **Metric**                               | **d = 0**    | **d = 1**    | **d = 2**     | **d = 3**     |
> |------------------------------------------|--------------|--------------|---------------|---------------|
> | # Forward Passes (can be parallelized)   | 1            | 9            | 25            | 49            |
> | Memory (CIFAR)                           | 114.50 MB    | 141.50 MB    | 182.93 MB     | 260.50 MB     |
> | Memory (ImageNet)                        | 157.80 MB    | 225.19 MB    | 384.03 MB     | 650.47 MB     |
>
> As shown, modern GPUs have ample memory headroom, and SR’s memory growth under normal inference remains modest. The more interesting observation arises during **adversarial attack generation** on one single 224$\times$224 inputs: memory usage increases dramatically, from **346.19 MB** for standard inference to **10649 MB** when performing gradient-based adaptive attacks through multi-branch SR. This exponential increase makes strong adaptive attacks (“WorstCase” in Table 1) EXTREMELY difficult in practice, highlighting an important defensive advantage of SR. We will incorporate this expanded compute and memory analysis in the updated manuscript.
>
> **W4**: Thank you for the suggestion. We will clarify this in the revised manuscript and will open-source the full implementation. In practice, SR is fully plug-and-play: we simply insert an SR wrapper around the encoder and leave the remainder of the architecture completely unchanged. This means there is no modification to downstream components such as correlation volumes, matching modules, recurrent update blocks, cost-volume construction, or refinement stages. Because all modern flow and stereo models contain an encoder, this minimal change makes SR universally applicable across architectures.
>
> **Scope and Related Works**: We will sharpen the positioning of our method. While the main paper includes a direct comparison on CIFAR-10 classification, we also evaluated TTE on optical flow and observed that SR significantly outperforms TTE in this setting as well. Which is labeled as “Output-Ensemble” in Figure 4. We will update the labels and clarify.

---

> ### Author Response · Authors · 2025-11-21
> **Thank you for your support! (2/2)**
>
> **Q1**: See W3.
>
> **Q2**: Thank you for the suggestion. A direct latency-matched comparison with TTE or feature-denoising methods is not entirely meaningful in our setting, because those approaches do not reach the same robustness levels as SR, as shown in the experiments, even when given equal or greater inference-time budgets. For test-time TTE specifically, SR is faster at every comparable setting, since TTE performs full forward passes throughout the whole network and therefore scales less efficiently. For example in the optical flow experiment, running d=3, SR costs 1.25 s/image while TTE runs at 5.45 s/image. We will clarify this distinction and add the relevant discussion in the revised manuscript.
>
> **Q3**: Empirically, we observe that SR with integer translations provides monotonic gains up to d=one-quarter of the model’s effective downsampling factor (e.g., ViT patch size or CNN downsampling).
>
> **Q4**: For the RAFT experiments, the same integer translation is applied to both frames. The SR operations, translation, inverse alignment, and aggregation, are all performed before the correlation stage, i.e., strictly within the encoder. So the model reduces to single forward pass after the encoder, which explains its efficiency compared to TTE (see Q3)
>
> **Q5**: Thank you for raising this point. Localized patch attacks are indeed an interesting setting, and they prompt deeper theoretical consideration. As discussed in theoretical analysis in the comment to all reviewers, the SR formulation works best under the assumption that perturbations induce approximately “uncorrelated’’ directions in latent space across transformed inputs. This assumption is less accurate for localized, high-magnitude corruptions for two reasons: (1) the perturbations are large and visually salient, so first-order linearization may no longer capture their effect; and (2) unlike PGD-style noise, rain, snow, or patches are not visually meaningless patterns. Models are likely trained on data with similar patterns, making the independence assumption weaker.
>
> Nevertheless, as shown in **W1**, we additionally evaluated SR under two localized corruption families: **DistractingDownpour** [Schmalfuß et al., ICCV’23] and **Detection Defenses** [Scheurer et al., WACV’24]. Under these settings, SR recovers **7.5%** of accuracy loss for snow, **3.7%** for rain, and **5.8%** for patch-based attacks. While the gains are smaller than those observed for PGD or FGSM (consistent with the reviewer’s intuition), SR still provides meaningful improvements.
>
> To better understand this behavior, we inspected the error modes and found that SR consistently reduces the EPE *within the corrupted region* (e.g., the location of the patch or particles). We conjecture that SR “smooths out’’ the corruption-induced artifacts by aggregating misaligned feature responses across translations, thereby mitigating the localized disruption. We will expand our discussion of this phenomenon in the revised manuscript.

---

> ### Comment · Reviewer_6Kns · 2025-11-26
>
> Thank you for the detailed response! I appreciated the additional theoretical analysis in response to reviewers YvRA and HimK, which are a valuable addition to the original paper.
>
> Regarding W1, my comment was targeted at testing more optical flow / scene flow models (other models than RAFT and PSMNet) rather than evaluating the defense effectiveness on additional attacks. As more architectures were evaluated in response to the other reviewers, adding more optical flow / scene flow models would be a desirable add-on but not strictly necessary.
> The provided results on the additional attacks are very insightful, and in my view, additionally strengthen the contribution of the paper. These results provide evidence that the proposed test time defense is even valuable in scenarios where the underlying theoretical assumptions do not fully hold anymore, and showcases the practical potential of this defense type, and are a valuable addition to the revised paper.
>
> Overall, the responses are confirming my view that the paper proposes a valuable and widely applicable defense against adversarial perturbations, whose value has been experimentally demonstrated by the authors across problem domains, adversarial attack types and model architectures.
>
> I have also read all other reviews and corresponding author responses, and find the responses convincing and detailed. The authors have run additional experiments where needed, and addressed the questions for more theoretical analysis. Most importantly,  the simplicity and flexibility of the proposed defense has been highlighted by all reviews. I would like to thank the authors for their excellent work, and continue to support the acceptance of this paper.

---

### Official Review · Reviewer_YvRA · 2025-10-31

**Soundness:** 4
**Presentation:** 3
**Contribution:** 4
**Rating:** 6
**Confidence:** 4

**Summary:**

This paper presents a novel test-time defense mechanism against adversarial attacks by leveraging the principle of stochastic resonance. The core idea of "combating noise with noise" is innovative. The method's key strengths are its training-free, architecture-agnostic, and attack-agnostic nature. The experimental validation is particularly compelling, demonstrating state-of-the-art robustness not only in image classification but also, for the first time, providing a viable test-time defense for dense prediction tasks like stereo matching and optical flow. The following are the modification suggestions.

**Strengths:**

1.The paper introduces a novel test-time defense based on stochastic resonance, creatively applying a classical signal processing principle to adversarial robustness. It is also among the first to extend such defenses to dense prediction tasks, opening a new research avenue.
2.The method is technically sound and well-validated through extensive experiments across datasets, architectures, and attack types. The consistent performance gains and ablation studies support the robustness and generality of the approach.
3.The paper is clearly structured and well-written, with intuitive explanations and well-designed figures that make the underlying ideas of stochastic resonance easy to understand.
4.The approach is training-free, architecture-agnostic, and broadly applicable, providing both practical robustness gains and conceptual insights that can inspire future research on stochastic mechanisms in machine learning.

**Weaknesses:**

1.The paper lacks a formal explanation of how stochastic resonance enhances robustness; adding a theoretical model linking perturbation strength to robustness would improve clarity.
2.The trade-off between robustness and inference time is not fully analyzed; quantitative results on computational cost would strengthen practicality claims.
3.The paper could elaborate on deployment challenges under strict computational constraints.

**Questions:**

1.Could the authors provide a clearer theoretical explanation of why stochastic resonance improves robustness, and how the resonance level ddd quantitatively relates to robustness gains?
2.How does the transformation ensemble preserve natural image statistics while disrupting adversarial noise—can this be demonstrated or formalized?

---

> ### Author Response · Authors · 2025-11-21
> **Thank you for your support!**
>
> **W1**: Thank you for the suggestion. See the additional theoretical analysis and we will update the paper.
>
> **W2**: In addition to the analysis above, the trade-off can be measured empirically, although we note that the specific trend depends on the task, the architecture, the evaluation metric, and the attack type. Below, we present a consolidated analysis using **ResNet-50** under different translation radii \(d\), illustrating how computational budget and adversarial robustness co-vary.
>
> | **Metric** | **d = 0** | **d = 1** | **d = 2** | **d = 3** |
> |-----------|-----------|-----------|-----------|-----------|
> | **# Forward Passes** | 1 | 9 | 25 | 49 |
> | **Wall-clock Time (s)** | 0.095 | 0.114 | 0.133 | 0.158 |
> | **SR Forward Time (excluding interpolation)** | 0.007 | 0.010 | 0.019 | 0.028 |
> | **Top-1 Acc @ ε = 8/255** | 8.51 | 18.40 | 20.01 | 21.17 |
> | **Top-1 Acc @ ε = 4/255** | 9.45 | 19.54 | 21.02 | 22.04 |
> | **Top-1 Acc @ ε = 2/255** | 11.10 | 21.02 | 22.44 | 23.44 |
> | **Top-1 Acc @ ε = 1/255** | 15.21 | 24.66 | 25.80 | 26.58 |
>
> Even at the most efficient setting $(d = 1)$, the improvement in robustness is substantial. Users can choose the desired operating point based on latency constraints and may also selectively sample transformations to further reduce computation. Notably, a significant portion of the overhead comes from translating images and performing interpolation via Python-level loops in PyTorch. The pure computation time of the network components under SR is reported as **“SR Forward Time,”** demonstrating that the method itself is efficient. Future work will focus on implementing these transformations in a custom CUDA kernel for further speedups (see **W3**).
>
> **W3**: Most of the computational overhead arises from data copying when generating the shifted augmentations, since these operations are performed inefficiently using a for-loop by `torchvision.transforms.affine` given the current standard PyTorch library. A custom CUDA kernel that performs the augmentations in-place would eliminate this bottleneck and drastically reduce the computational demand upon deployment. This is a natural direction for future research.
>
> **Q1**: See Additional Theoretical Analysis in official comment to all reviewers.
>
> **Q2**: Averaging with respect to a group of transformations is a generalized form of spatial anti-aliasing, where the kernel (say, a Gabor filter bank) is replaced by the trained embedding. Since the trained model already undergoes implicit forms of spatial anti-aliasing through network design, data augmentation, and regularization, local anti-aliasing operations do not alter the activation maps. Adversarial perturbations, on the other hand, are sample-specific and their statistics are space-dependent; such dependency is destroyed by the anti-aliasing operation.

---

### Author Response · Authors · 2025-11-21
**Additional Theoretical Analysis**

Thanks to the reviewers for the constructive comments. As requested by reviewers `YvRA` and `HimK`, we provide additional theoretical analysis for our method. We will provide the revised manuscript towards the end of the author-reviewer discussion period. Note that, the theoretical foundations of Stochastic Resonance have been long established in statistical physics, summarised for instance in https://journals.aps.org/rmp/abstract/10.1103/RevModPhys.70.223
. Our work applies this concept specifically to trained embeddings.

In the context of adversarial perturbations, modern deep networks often exhibit poor local Lipschitz regularity, where small input perturbations induce disproportionately large latent-space changes, exploited by adversaries.

If $x$ is a clean signal (an image in our case) and $\tilde{x} = x + n(x)$ its adversarially perturbed counterpart. Let $\varphi$ denote the encoder up to the SR layer. Using a discretized set of translations $ \lbrace g_i \rbrace^N_{i=1} \subset G $ and their corresponding inverse $ g^{-1}_i$, the SR feature is:

$$
\hat{\varphi}(x)=\frac{1}{N}\sum_{i=1}^N g^{-1}\_{i}\varphi(g_i x), \qquad \hat{\varphi}(\tilde{x})=\frac{1}{N} \sum_{i=1}^N g^{-1}_{i}\varphi(g_i \tilde{x})
$$

The first-order approximation yields:

$$
\varphi(g_i \tilde{x}) = \varphi(g_i x + g_i n(x))
\approx \varphi(g_i x) + J_{g_i x}[g_i n(x)],
$$

where $J_{g_i x}$ is the Jacobian of $\varphi$ at $g_i x$. We have

$$
\Delta_{\mathrm{SR}}(x)
=\hat{\varphi}(\tilde{x})-\hat{\varphi}(x)
\approx \frac{1}{N}\sum_{i=1}^N g^{-1}_{i} J\_{g_i x} [g_i n(x)].
$$

In contrast, a standard encoder responds as:

$$
\Delta_{\mathrm{base}}(x)
=\varphi(\tilde{x})-\varphi(x)
\approx J_x n(x),
$$

which is dominated by the local worst-case Lipschitz direction at $x$.

The terms $g^{-1}\_{i}J\_{g_i x}[g_i n(x)]$ in the SR expansion are generally not aligned across $i$:
(1) the adversarial noise $n(x)$ is crafted for a single input $x$, so under translation it is re-aliased into different patches;
(2) the Jacobian $J_{g_i x}$ varies across $i$, so the maximally-amplifying direction at $x$ is not simultaneously amplifying for all transformed inputs;
(3) the signal terms $\varphi(g_i x)$ align under the inverse warp, while the Jacobian–noise terms do not.

Under mild decorrelation assumptions:

$$
\left|\frac{1}{N}\sum_{i=1}^N v_i\right|^2
\sim \frac{1}{N}\mathbb{E}[v_i^2],
\qquad
v_i=g^{-1}\_{i}J\_{g_i x}[g_i n(x)],
$$

so the effective Lipschitz constant of the SR map decreases by approximately $1/\sqrt{N} \sim 1/d$ where $d$ is the SR level in our paper. Thus, SR suppresses adversarially amplified latent-space outliers while preserving coherent scene structure.

This analysis is also validated empirically. Using a pre-trained DINO encoder, we add adversarial perturbations that maximally distort the latent embedding, and then apply SR with increasing translation radii $d$. The $L_1$ distance between adversarial and clean embeddings decreases as:

d = 0 (no SR defense): $L_1=12.30$

d = 1: $L_1=5.29$

d = 2: $L_1=3.50$

d = 3: $L_1=2.64$

d = 4: $L_1=2.12$

which closely follows the expected $1/\sqrt{N}$ decay pattern. A complete analysis of SR is beyond the scope of this paper but hopefully this analysis helps reassure the reader that the framework is sound.

---

### Author Response · Authors · 2025-12-03

Dear Area Chair and Reviewers,

We sincerely thank you for your time, thoughtful feedback, and support throughout the review process. Although the discussion period was unexpectedly cut short, we made every effort to address all concerns raised. Because reviewers were unable to finalize their evaluations, we provide the following for your reference regarding the additions and revisions made during rebuttal.

During the discussion phase, we responded in detail to each question and substantially strengthened both the theoretical and empirical components of the paper. In particular:

- As suggested by YvRA and HimK, we added a new theoretical analysis explaining why stochastic resonance improves robustness, including a first-order expansion and a Lipschitz-reduction argument, and validated these insights with empirical measurements using the DINO ViT encoder. This was also explicitly appreciated by 6Kns and 2PRW during the discussion.
- We incorporated additional evaluations using AutoAttack as suggested by 6Kns, showing that our method achieves the strongest robust accuracy across all baselines.
- Also suggested by 6Kns, we evaluated our method on CIFAR-10-C, covering 19 real-world corruption types. SR recovers an average of 20.8% of lost performance, with more than 30% recovery for blur, fog, and saturation corruptions.
- To answer 6Kns's question regarding localized attacks, we expanded our optical-flow experiments to include adverse-weather perturbations (snow and rain), recovering 7.5% and 3.7% of accuracy loss, respectively.
- In response to YvRA and 6Kns, we provided a more comprehensive speed and memory analysis, including sequential and parallel timing, memory scaling with SR depth, and the substantial memory overhead required for adaptive attacks, highlighting SR's practical defensive advantage.
- Per the request of HimK on domain adaptation, we conducted a sim-to-real, cross-resolution experiment using DepthAnything V2 trained on VKITTI and tested on KITTI, where SR provides consistent improvements across all metrics.

These additions (and other minor points) have addressed all the concerns/questions raised by the reviewers. We have integrated them into the revised manuscript, reflecting the expanded theory, new experiments, compute analysis, and the additional citations suggested by the reviewers.

We greatly appreciate the constructive feedback and the time invested by both the reviewers and the Area Chair. We hope that the substantial theoretical and experimental progress made during the rebuttal, along with our comprehensive responses, will be taken into consideration in the final evaluation.

Submission #4169 Authors

---

### Meta-Review · Area_Chair_TUFV · 2026-01-17

**Summary:**

This paper presents a test-time defense mechanism against adversarial attacks by leveraging the principle of stochastic resonance.

Reviewers expressed the following concerns.
- The paper lacks a formal explanation of how stochastic resonance enhances robustness
- Computational complexity and trade off between robustness and inference time require discussion
- Experiments for stereo and optical flow are limited to one model.
- While the introduction mentions general group actions, the method and experiments are limited to translations only. For rotations, robustness degrades.
- The original version of the paper did not include experiments with AutoAttack

**Reviewer Concerns:**

Authors provided responses to most of the reviewer concerns.

- The main concerns about theoretical analysis to show effectiveness of the proposed approach remain outstanding. While authors provided a brief theoretical analysis and discussion, it leaves the main concerns largely unaddressed. The bounds require additional explanation and verification. The analysis seems to assume linearity and approximation in the decomposition, which are not clearly explained.

- Authors included multiple experiments to highlight effectiveness of the proposed method on different images and models.

**Reviewer Scores:**

Reviewer scores: 6, 8, 4, 4

- The main outstanding concern in my view is that the paper is missing a convincing theoretical analysis on how and when stochastic resonance will be effective for test-time defense.
- It is unclear which group actions will be effective for different inference tasks. While classification methods can remain robust to translation, such a step may adversely affect the performance of other dense prediction tasks (e.g., segmentation and object detection).

I think these changes will require a careful review.

---

### Decision · Program_Chairs · 2026-01-26

Reject